# LIF Recurrent Memory Enables Long-Horizon Spiking Computation

**Fenghao Liu** [1]  **Yipeng Shen** [1]  **Peng Chen** [1 2]  **Qian Zheng** [1]  **Peng Lin** [1 2]  **Gang Pan** [1]

## Abstract

Processing long sequence data such as speech requires models to maintain long-term dependencies, which is challenging for recurrent spiking neural networks due to high temporal dynamics in neuron models that leak stored information in their membrane potentials, and due to vanishing gradients during backpropagation through time. These issues can be mitigated by employing more complex neuron designs, such as ALIF and TC-LIF, but these neuron-level solutions often incur high computational costs and complicate hardware implementation, undermining the efficiency advantages of spiking neural networks. Here we propose an architectural-level solution that leverages the dynamical interactions of a few leaky integrate-and-fire (LIF) neurons to enhance long-term information storage. The memory capability of this LIF-based micro-circuit is adaptively modulated by global recurrent connections of the recurrent spiking neural network, contributing to selective enhancement of temporal information retention, and promoting stable gradient propagation through time. The proposed model outperforms baselines including LSTM, ALIF, and TC-LIF in long sequence tasks, achieving 96.52% accuracy on the PS-MNIST dataset. Furthermore, our method also provides a compelling efficiency advantage, yielding up to 277× computational efficiency improvement compared to conventional models such as LSTM. This work paves the way for building cost-effective, hardware-friendly, and interpretable spiking neural networks for long sequence modeling.

[1]The State Key Lab of Brain-Machine Intelligence, College of Computer Science and Technology, Zhejiang University, Hangzhou, Zhejiang, China [2]Nanhu Brain-Computer Interface Institute, Hangzhou, Zhejiang, China. Correspondence to: Peng Lin <penglin@zju.edu.cn>.

*Proceedings of the 43rd International Conference on Machine Learning*, Seoul, South Korea. PMLR 306, 2026. Copyright 2026 by the author(s).

## 1. Introduction

Spiking neural networks (SNNs) offer energy-efficient computing paradigms by leveraging brain-inspired neuron models as activation functions to enable sparse and event-driven computations (Roy et al., 2019). The leaky integrate-and-fire (LIF) neuron is the most widely adopted neuron model in SNNs, which integrates input signals and generates a spike once the membrane potential exceeds its firing threshold (Gerstner & Kistler, 2002). To enhance temporal resolution for sequential inputs, the LIF neuron incorporates a leak mechanism that effectively filters out irrelevant long-term information, making SNNs a good candidate for temporal signal processing tasks using recurrent spiking neural network (RSNN) architectures (Bellec et al., 2018).

Nonetheless, the performance of LIF-based RSNNs, particularly in long-sequence modeling, still faces three major challenges: (1) the leak mechanism, while beneficial for short-term dynamics, causes the LIF neuron to forget earlier inputs, hindering the capture of long-term dependencies; (2) RSNNs with simple recurrent connections lack adaptive mechanisms to dynamically regulate information flow based on input salience, making them ineffective at distinguishing useful information from noise; (3) training RSNNs via backpropagation through time (BPTT) (Werbos, 2002) is impeded by the vanishing gradient problem, which greatly limits the model's overall performance.

To overcome the short-term memory limits of the vanilla LIF neuron, several complex neuron models have been proposed to incorporate additional mechanisms such as adaptive thresholds (Bellec et al., 2018), compartmental dynamics (Zhang et al., 2024a), or variable time constants (Fang et al., 2021a) in individual spiking neurons. Although these approaches have demonstrated improved robustness in long-sequence modeling, their model complexity leads to a high computational cost and additional design overhead for neuromorphic hardware.

Rather than relying on the intrinsic properties of individual neurons for long-term memory, an alternative approach is to leverage the collective dynamics at the architectural level. For example, long short-term memory (LSTM) networks in artificial neural networks (ANNs) address the long-sequence problem by introducing a gated cell state. However, the gating mechanism is not natively supported by most neuro-

morphic processors, since many neuromorphic processors, such as Loihi 2 (Abreu et al., 2025), rely on specialized arithmetic logic units (ALUs) to emulate neuron operations which only support low-precision, element-wise operations such as integer and fixed-point addition, multiplication, and bitwise shifts, which imposes inherent limitations when approximating nonlinear functions such as tanh, sigmoid or exponential functions. Directly performing these non-native operations using these ALUs might suffer significant accuracy loss and require support from external high precision units.

In this work, we propose a novel recurrent architecture of RSNN for long sequence modeling, based on hierarchical recurrent connections including a compact local LIF recurrent memory module (LRMM). The LRMM uses four vanilla LIF neurons, forming an input pathway, an output pathway, and a local recurrent memory loop that dynamically regulates the stored information without gating units, while offering native compatibility with neuromorphic hardware. The local memory loop also enhances gradient propagation under BPTT with reduced vanishing gradients, demonstrating stable gradient retention for training of RSNNs. We evaluated our model on several long-sequence benchmarks, including PS-MNIST, SHD, SSC, and Binary Adding task. Our approach outperforms standard LIF networks and complex neuron-centric models such as TC-LIF in terms of accuracy, gradient stability, and robustness to long sequences. It also demonstrated performance comparable to complex architectures such as LSTM, while maintaining excellent computing efficiency with $277\times$ lower energy than LSTM. Our contributions are summarized as follows:

- We design a vanilla LIF based recurrent memory module that incorporates a local memory loop for long-term information retention without complex neuron designs.

- We employ a hierarchical recurrent architecture that combines global recurrent connections and local recurrent memory to dynamically and selectively regulate the memory of input data.

- We show that the memory loop improves gradient propagation under BPTT and enhances the gradient retention factor, thereby mitigating the vanishing gradient problem in training.

- We validate our model on four long-sequence benchmarks (PS-MNIST, SHD, SSC, and Binary Adding task), demonstrating improved accuracy, stable training dynamics, and superior energy efficiency.

## 2. Related Work

**Long-Term Memory in SNNs.** A key challenge in SNNs is retaining information over a long time. Several neuron-

centric approaches address this issue by modifying LIF dynamics, such as adaptive thresholds in ALIF (Bellec et al., 2018), radial dynamics in RadLIF (Bittar & Garner, 2022), and dual-compartment coupling in TC-LIF (Zhang et al., 2024a). Although effective, they increase model complexity and require hardware-specific tuning, limiting their efficiency and scalability.

**Gated Recurrent Models.** Recurrent architectures such as LSTM (Hochreiter & Schmidhuber, 1997) and GRU (Cho et al., 2014) achieve strong performance in sequential tasks by explicitly gating and storing information over time (Datta et al., 2022). However, both conventional and spiking counterparts, such as Spiking-LSTM (Lotfi Rezaabad & Vishwanath, 2020), rely on complex gating and expensive state updates, which limit their suitability for neuromorphic computing.

**Structural Complexity in SNNs.** Another line of work enhances temporal processing in SNNs by introducing architectural complexity, such as locally recurrent motifs (Zhang et al., 2024b), small-world connectivity (Pan et al., 2024), and brain-inspired topologies (Wang et al., 2024). These studies suggest that structural complexity can benefit temporal modeling in SNNs, yet they do not provide explicit mechanisms to sustain long-term dependencies. Spiking LMU-Former (Liu et al., 2024) introduces long-range memory through state-space models, but its deep convolutional architecture leads to high computational and parameter costs, making it less suitable for neuromorphic hardware.

**Event-driven and State-space Sequence Models.** Recent event-driven and state-space models, such as EGRU (Subramoney et al., 2022) and Event-SSM (Schöne et al., 2024), also provide strong temporal modeling baselines for sequential data. These methods improve long-range sequence modeling through either more complex recurrent update rules or dense state-space computations. Similarly, these models also leads to high computational costs.

## 3. Method

### 3.1. Vanilla LIF Based Recurrent Memory Module

**Spiking Neuron Model.** We employ the vanilla LIF neuron as the fundamental computational unit in our RSNNs. The membrane potential $u(t)$ evolves over time according to the following differential equation:

$$\tau \frac{du(t)}{dt} = -(u(t) - u_{reset}) + RI(t). \qquad (1)$$

Here, $\tau$ is the membrane time constant, $R$ is the resistance, $I(t)$ is the synaptic input, and $u_{\text{reset}}$ is the reset potential. A spike $S(t)$ is emitted when the membrane potential exceeds the threshold $V_{th}$, after which it is reset to $u_{\text{reset}}$. For practical implementation, we discretize the equation using

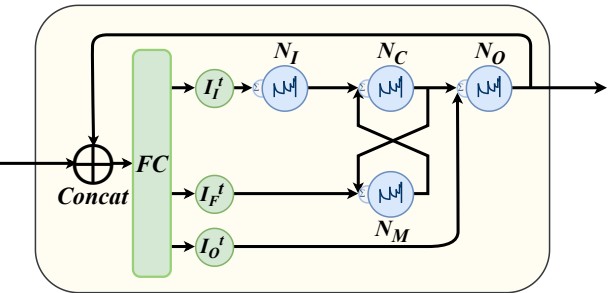

*Figure 1.* Vanilla LIF based recurrent memory module framework.

the Euler method. Assuming $u_{reset} = 0$ and $R = \tau$, the discrete-time update with soft reset is:

$$u[t+1] = \left(1 - \frac{1}{\tau}\right)(u[t] - V_{th} S[t]) + I[t], \quad (2)$$

$$S[t] = \Theta(u[t] - V_{th}). \quad (3)$$

Here, $\Theta(\cdot)$ is the Heaviside step function, which outputs 1 when its argument is positive and 0 otherwise. This prevents runaway spiking while preserving the residual subthreshold voltage, and the leak factor $(1-\frac{1}{\tau})$ still governs the temporal decay between spikes.

**Recurrent Memory Module Design Based on LIF Neurons.** We propose a lightweight and interpretable recurrent memory module composed entirely of vanilla LIF neurons, as shown in Figure 1. Each module contains four LIF units with distinct functional roles, i.e., input integration neuron $N_I$ that processes incoming and global feedback signals, memory neuron $N_M$ that maintains long-term temporal memory, context neuron $N_C$ that combines the memory and the input, and output control neuron $N_O$ that determines the readout information. The LRMM module is plugged into the global recurrent architecture of RSNN, which receives three modulatory currents, i.e., $I_I, I_F, I_O$ that dynamically regulate the information flow based on input salience. These structured global and local interactions enable the LRMM module to retain long-term information within a fully spike-driven and biologically plausible framework. Each of the three input currents $j \in \{I, F, O\}$ is computed from the same combination of the current input $I[t]$ and the previous output spike $S_O[t-1]$, using separate fully connected layers.

$$I_j[t] = \Phi\Big(W_j\,[\,I[t]; S_O[t-1]\,]+b_j\Big), \quad j \in \{I, F, O\}, \quad (4)$$

where $[\,\cdot\,;\,\cdot\,]$ denotes feature-wise concatenation between the current input $I[t]$ and the previous output spike $S_O[t-1]$. The modulation function $\Phi$ interpolates between the standard sigmoid and a piecewise-linear (PL) hard-sigmoid:

$$\Phi(z) = (1-m)\,\sigma(z) + m\,\mathrm{PL}(z), m \in [0, 1], \quad (5)$$

$$\mathrm{PL}(z) = \mathrm{clip}\Big(0.5 + \frac{z}{2a}, 0, 1\Big), a > 0, \quad (6)$$

with $\mathrm{clip}(x,0,1) = \min(\max(x,0),1)$. During training, we anneal $m$ from 0 to 1. At inference, we set $m = 1$ so that $I_j[t] = \mathrm{PL}(z)$, which provides a hardware-friendly approximation. At inference, Eq. (4)–(6) are implemented as one linear transformation followed by a piecewise-linear clipping operation. The modulation is applied per LRMM cell to its own local loop, rather than as a global controller shared across cells.

The inputs of four neurons in the LRMM module can be formulated as:

$$I_{N_I}[t] = k_I \cdot I_I[t], \quad (7)$$

$$I_{N_M}[t] = w_{C,M} \cdot S_C[t-1] + k_F \cdot I_F[t], \quad (8)$$

$$I_{N_C}[t] = w_{I,C} \cdot S_I[t] + w_{M,C} \cdot S_M[t], \quad (9)$$

$$I_{N_O}[t] = w_{C,O} \cdot S_C[t] + k_O \cdot I_O[t]. \quad (10)$$

where $S_X[t]$ denotes the spike output of neuron $N_X$ at time $t$ and $I_{N_X}[t]$ denotes its corresponding input current. $k_X \in \mathbb{R}$ means the trainable scaling factor for input neurons. The weight $w_{a,b} \in \mathbb{R}$ specifies the synaptic connection from neuron $a$ to neuron $b$. All parameters $\{W_j, b_j\}_{j \in \{I,F,O\}}$ in Eq.(4) and all scaling factor $k_X$ and all synaptic scalars $w_{a,b}$ in Eq.(7) to Eq. (10) are learnable and time-shared across $t$. This schedule ensures causal updates and avoids algebraic loops, since $N_M$ depends on $S_C[t-1]$ while $N_C$ consumes the freshly produced $S_I[t]$ and $S_M[t]$.

The proposed LRMM enables effective temporal modeling by exploiting dynamic interactions among event-driven LIF neurons. Unlike approaches that rely on complex neuron-level modifications, our method provides an architectural level solution that supports both short-term modulation and long-term memory integration within a fully spike-based architecture. The module introduces adaptive regulation of input salience, allowing it to selectively determine which information is integrated, stored, and read out. The use of vanilla LIF and compact RSNN architecture enhances runtime efficiency and compatibility with neuromorphic hardware implementations.

### 3.2. Stability Analysis of Backpropagation Through Time (BPTT)

In this subsection, we analyze the BPTT dynamics using the surrogate computation graph during training. The **temporal** gradient component from membrane leakage and reset is decoupled from the **spatial** gradient component of reverse-mode accumulation along spike-to-current pathways. The separated pathways make the sources of gradient amplification and attenuation explicit, enabling a principled stability analysis. For clarity, Figure 2 illustrates the forward information flow in the proposed recurrent memory module. We further investigate how local recurrent loops contribute to stable long-range gradient propagation across extended temporal horizons.

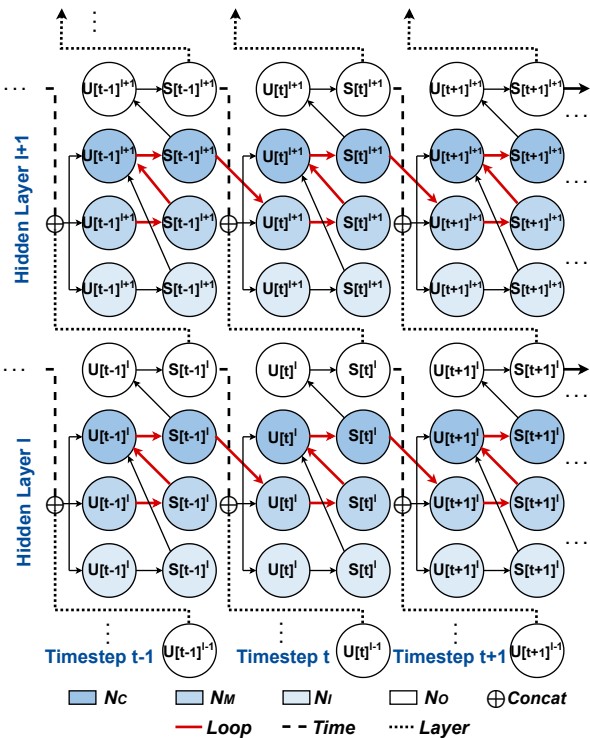

*Figure 2.* Forward information flow in the proposed recurrent memory module. Red bold arrows indicate temporal feed-forward paths between $N_M$ and $N_C$. Dashed arrows indicate temporal and layer-wise forward connections. Note that during BPTT, gradients propagate in reverse direction of the arrows.

**Definition.** We index time $t$ and neuron $N_X$, $X \in \{I, M, C, O\}$ and current gates $j \in \{I, F, O\}$. Each unit keeps membrane $U_X[t]$, spike $S_X[t]$, input $I_X[t]$, leak $\alpha_X \in (0, 1)$. We reuse $gZ[t] = \partial\mathcal{L}/\partial Z[t]$ for any symbol $Z$ (e.g., $gU_X[t], gS_X[t], gI_j[t]$) and parameter gradients $\partial\mathcal{L}/\partial w$. We further define $A_X[t]$ to denote the one-step temporal gain along the local membrane update (leak and reset) with $I_X[t+1]$ fixed, defined as:

$$A_X[t] := \left.\frac{\partial U_X[t+1]}{\partial U_X[t]}\right|_{I_X[t+1] \text{ fixed}} \quad (11)$$
$$= \alpha_X\left(1 - V_{\text{th}}\,\sigma'_X(U_X[t] - V_{\text{th}})\right).$$

where $A_X[t]$ is the one-step temporal gain of neuron $X$. Inter-step dependencies mediated through synaptic currents are handled separately via the spatial reverse-mode paths. It measures how membrane leak and threshold reset contract or expand the mapping along time, with $A_X[t] \approx \alpha_X$ when $\sigma'_X \approx 0$ far from threshold and a smaller value near threshold due to the subtractive reset.

**Per-neuron One-step Temporal Recursion.** At time $t$, $U_X[t+1]$ depends on $U_X[t]$ (leak and reset) and exogenous $I_X[t+1]$; $S_X[t]$ depends on $U_X[t]$. We keep forward firing

hard and use a smooth surrogate only in back propagation:

$$S_X[t] = \sigma_X(U_X[t] - V_{\text{th}}). \quad (12)$$

Using the soft-reset update and Eq.(11), the one-step temporal gain is $A_X[t]$ as defined above. With the boxcar surrogate in A.1 and $H_w = V_{\text{th}}/2$, $V_{\text{th}} = 1$ , $\sigma'_X(\cdot) \in \{1, 0\}$ and $A_X[t] \in \{\alpha_X, 0\}$, thus

$$A_X[t] = \alpha_X \text{ if } |U_X[t] - V_{\text{th}}| > H_w,$$
$$\text{and } A_X[t] = 0 \text{ otherwise.} \quad (13)$$

By the chain rule, the loss gradient to $U_X[t]$ decomposes into a temporal path and a spike path:

$$gU_X[t] = \underbrace{A_X[t]\,gU_X[t+1]}_{\text{temporal}} + \underbrace{\sigma'_X(U_X[t] - V_{\text{th}})\,gS_X[t]}_{\text{spatial}}. \quad (14)$$

**Per-neuron One-step BPTT Recursions.** By Eq. (14), letting $\sigma'_X[t] \equiv \sigma'_X(U_X[t] - V_{\text{th}})$ and using $A_X[t]$ from Eq. (11), the membrane adjoint of each neuron admits a one-step BPTT recursion. Specifically, $gU_X[t]$ decomposes into (i) a temporal contribution propagated from its own next-step adjoint $gU_X[t+1]$ scaled by the local temporal gain $A_X[t]$, and (ii) spatial cross-unit contributions routed through spike-to-current pathways, which inject gradients from other neurons at time $t$ and $t+1$ depending on the forward dependencies. The resulting per-neuron recursions are summarized below (see Appendix A.1 for details):

$$gU_I[t] = A_I[t]\,gU_I[t+1] + \sigma'_I[t]\,w_{I,C}\,gU_C[t], \quad (15)$$
$$gU_M[t] = A_M[t]\,gU_M[t+1] + \sigma'_M[t]\,w_{M,C}\,gU_C[t], \quad (16)$$

$$gU_C[t] = A_C[t]\,gU_C[t+1]$$
$$+ \sigma'_C[t]\Big(w_{C,O}\,gU_O[t] + w_{C,M}\,gU_M[t+1]\Big). \quad (17)$$

**Loop-Induced Effective Temporal Gain in $N_M$−$N_C$.** We analyze how the $N_M$−$N_C$ loop affects the one-step temporal operator in BPTT. Starting from the per-unit adjoint recursions, we define the loop couplings $\beta_t$ and $\gamma_t$ and apply a single substitution. From the per-neuron BPTT recursions of $N_C$ and $N_M$ in Eqs. (16) and (17), we define

$$\beta_t := \sigma'_C[t]\,w_{C,M}, \qquad \gamma_t := \sigma'_M[t]\,w_{M,C}. \quad (18)$$

Substituting $gU_M[t+1] = A_M[t+1]gU_M[t+2] + \gamma_{t+1}gU_C[t+1]$ into $gU_C[t]$ yields

$$gU_C[t] = \underbrace{(A_C[t] + \beta_t\gamma_{t+1})}_{\text{effective temporal gain}} gU_C[t+1]$$
$$+ \beta_t A_M[t+1]\,gU_M[t+2] + \sigma'_C[t]\,w_{C,O}\,gU_O[t]. \quad (19)$$

$$gU_M[t] = \underbrace{(A_M[t] + \gamma_t\beta_t)}_{\text{effective temporal gain}} gU_M[t+1]$$
$$+ \gamma_t A_C[t] \, gU_C[t+1] + \gamma_t \, \sigma'_C[t] \, w_{C,O} \, gU_O[t]. \tag{20}$$

Equations (19) and (20) yield direct one-step recursions $gU_C[t+1] \to gU_C[t]$ and $gU_M[t+1] \to gU_M[t]$ with effective temporal gains $A_C[t] + \beta_t\gamma_{t+1}$ and $A_M[t] + \gamma_t\beta_t$, respectively. Compared with the leak-only LIF baseline where the gains equal $A_C[t]$ and $A_M[t]$, the loop contributes an additional coupling term $\beta_t\gamma_{t+1}$ or $\gamma_t\beta_t$, which establishes a direct pass-through across consecutive steps and reduces reliance on the leak factor $A_X[t]$.

Recall $A_X[t] = \alpha_X(1 - \sigma'_X[t])$ from Eq.(11), and define

$$G_C[t] := A_C[t] + \beta_t\gamma_{t+1}$$
$$= \alpha_C(1 - \sigma'_C[t]) + \sigma'_C[t] \, \sigma'_M[t+1] \, w_{C,M} \, w_{M,C},$$
$$G_M[t] := A_M[t] + \gamma_t\beta_t$$
$$= \alpha_M(1 - \sigma'_M[t]) + \sigma'_M[t] \, \sigma'_C[t] \, w_{M,C} \, w_{C,M}. \tag{21}$$

The quantities $G_C[t]$ and $G_M[t]$ comprise two complementary components that are active in different operating regimes. For $G_C[t]$ one has

$$G_C[t] = \underbrace{\alpha_C(1 - \sigma'_C[t])}_{\text{off-threshold contribution}} + \underbrace{\sigma'_C[t] \, \sigma'_M[t+1] \, w_{C,M} \, w_{M,C}}_{\text{near-threshold loop contribution}}$$

and for $G_M[t]$ one has an analogous decomposition. Thus each gain contains an off-threshold term proportional to $(1 - \sigma')$ and a near-threshold loop term proportional to $\sigma'$. Rather than implying a pointwise guarantee of nonzero gain at every step, this structure reduces the occurrence and length of zero-gain (or near-zero-gain) segments by providing an additional pass-through whenever consecutive near-threshold events arise on the loop.

In particular, when $\sigma'_C[t] \approx 1$ and $\sigma'_M[t+1] \approx 1$ (or symmetrically $\sigma'_M[t] \approx 1$ and $\sigma'_C[t] \approx 1$), the loop contribution dominates and conveys gradients through the interconnecting synapses with magnitude controlled by $w_{C,M}w_{M,C}$. This mitigates gradient disconnection around spike-adjacent time steps and reduces reliance on the leak factor alone.

## 4. Experiments

### 4.1. Experimental Setup

**Datasets.** We evaluate our models on four widely used benchmark datasets for sequential and event-driven learning, i.e., SHD (Cramer et al., 2020), SSC (Warden, 2018), PS-MNIST (Le et al., 2015), and Binary Adding (Ma et al., 2025). These datasets are chosen to cover a diverse range of temporal modeling challenges, including event-based auditory processing, speech command recognition, long-range dependency reasoning, and numerical sequence addition.

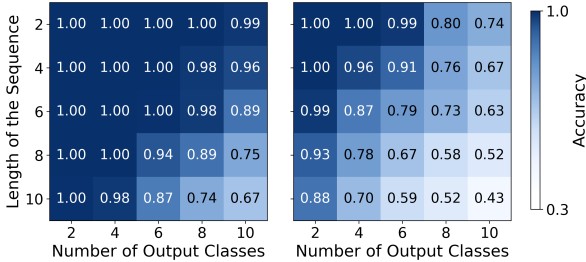

Figure 3. Copy-task accuracy under a fixed delay of $delay_t = 20$, across sequence length $L$ and alphabet size $K$.

A detailed description of each dataset is provided in Appendix A.2.

**Models.** Sequential inputs are fed directly into the network without spike encoding. The recurrent computations are handled by our proposed **LIF-based recurrent memory module (LRMM)**, which uses structured recurrent connections among LIF neurons to support memory formation in a fully spike-driven and biologically plausible manner. This recurrent design enables stable gradient propagation, long-term temporal integration, and selective retention of salient input patterns. By default, we use a two-layer LRMM backbone with 128 units per layer, followed by a linear classifier on the final hidden state.

**Training Details.** All training configurations, including hyperparameters and optimization strategies, are provided in Appendix A.3.

**Baseline Models and Comparative Methodology.** Detailed baseline configurations and comparison settings are provided in Appendix A.4.

### 4.2. Main Results

We present a comprehensive evaluation of LRMM to demonstrate its effectiveness across four dimensions, i.e., accuracy, gradient behavior, memory capability, and gating selectivity in long horizon temporal classification. **First**, on standard benchmarks including PS-MNIST, SHD, SSC, and Binary Adding, LRMM achieves strong and competitive accuracy with comparable or reduced parameter counts under a unified training protocol. **Second**, on long-range dependency settings, LRMM maintains accurate retrieval despite extended delays and noise, demonstrating robust long-term memory. **Third**, analysis of circuit-level activity shows that the recurrent loop adaptively modulates memory traces, enhancing informative segments and suppressing irrelevant ones. Causal interventions further demonstrate the necessity of this adaptive recurrence for long-term retention. **Fourth**, analysis of gradient flow shows that LRMM mitigates vanishing gradients and preserves better temporal credit assignment over long horizons, as quantified by the Gradient

*Table 1.* **Results on Temporal Benchmarks.** Evaluation on PS-MNIST, SSC, SHD, and Binary Adding task, reporting accuracy (%) and parameter counts. **LRMM** employs only vanilla LIF neurons, while **LRMM-ALIF** replaces $N_I$ and $N_O$ with ALIF neurons.

| Datasets | Method | Recurrent | Vanilla LIF | Parameters | Accuracy (%) |
|---|---|---|---|---|---|
| PS-MNIST (T=784) | LIF(Zhang et al., 2024a) | Y | Y | 0.155M | 80.39 |
| | LSTM (Rusch & Mishra, 2021) | Y | N | 0.27M | 92.90 |
| | GLIF (Yao et al., 2022) | Y | N | 0.15M | 90.47 |
| | ALIF (Yin et al., 2021) | Y | N | 0.15M | 94.30 |
| | BRFN (Higuchi et al., 2024) | N | N | 0.068M | 95.20 |
| | TC-LIF (Zhang et al., 2024a) | Y | N | 0.063M/0.15M | 92.69 / 95.36 |
| | **LRMM (ours)** | Y | Y | 0.15M | **96.52** |
| | **LRMM-ALIF (ours)** | Y | N | 0.15M | **97.39** |
| SSC (T=100) | LIF (Cramer et al., 2020) | Y | Y | 0.11M | 50.90 |
| | TC-LIF (Zhang et al., 2024a) | Y | N | 0.11M | 61.90 |
| | LSTM (Cramer et al., 2020) | Y | N | 0.43M | 73.10 |
| | SNN-CNN (Sadovsky et al., 2023) | N | N | N/A | 72.03 |
| | ALIF (Yin et al., 2021) | Y | N | N/A | 74.20 |
| | SpikGRU (Dampfhoffer et al., 2022) | Y | N | 0.28M | 77.00 |
| | RadLIF (Bittar & Garner, 2022) | N | N | 3.9M | 77.40 |
| | **LRMM (ours)** | Y | Y | 0.20M | **79.75** |
| | **LRMM-ALIF (ours)** | Y | N | 0.20M | **80.51** |
| SHD (T=100) | LIF (Cramer et al., 2020) | Y | Y | 0.108M | 71.40 |
| | LSTM (Cramer et al., 2020) | Y | N | 0.43M | 89.20 |
| | TC-LIF (Zhang et al., 2024a) | Y | N | 0.15M | 88.91 |
| | ALIF (Yin et al., 2021) | Y | N | N/A | 90.40 |
| | RadLIF (Bittar & Garner, 2022) | Y | N | 3.9M | 94.62 |
| | **LRMM (ours)** | Y | Y | 0.42M | **94.70** |
| | **LRMM-ALIF (ours)** | Y | N | 0.42M | **95.32** |
| Binary Adding (T=100) | LIF (Ma et al., 2025) | N | Y | 0.04M | 53.35 |
| | PLIF (Fang et al., 2021b) | Y | N | N/A | 53.25 |
| | adLIF (Bellec et al., 2018) | Y | N | N/A | 68.00 |
| | ALIF (Yin et al., 2021) | Y | N | N/A | 99.05 |
| | GLIF (Yao et al., 2022) | Y | N | N/A | 63.60 |
| | TC-LIF (Zhang et al., 2024a) | Y | N | N/A | 19.90 |
| | LM-H (Hao et al., 2023) | Y | N | N/A | 96.10 |
| | CLIF (Huang et al., 2024) | Y | N | N/A | 64.30 |
| | DH-LIF (Zheng et al., 2024) | Y | N | N/A | 99.35 |
| | **LRMM (ours)** | Y | Y | 0.15M | **99.55** |
| | **LRMM-ALIF (ours)** | Y | N | 0.15M | **100.00** |

Retention Factor.

**Results on Temporal Benchmarks.** Under reported baseline settings and comparable parameter regimes where available, LRMM achieves strong accuracy with compact model sizes. On PS-MNIST, LRMM achieves 96.52% with 0.15M parameters, exceeding TC-LIF at 95.36% with 0.15M. On SSC, LRMM reaches 79.75% with 0.20M, outperforming RadLIF at 77.40% with 3.9M. On SHD, LRMM obtains 94.70% with 0.42M compared with 94.62% for RadLIF with 3.9M. On Binary Adding, LRMM records 99.55% with 0.15M, slightly higher than 99.35% for DH-LIF. These results show that our simple recurrent LIF-based circuit matches or outperforms deeper and more complex models on long sequence tasks. These results, summarized in Table 1, indicate consistent improvements at compact model

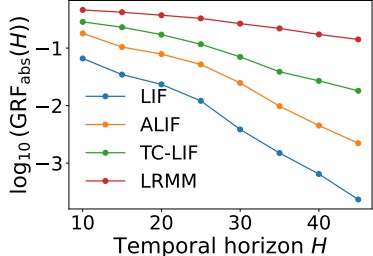

*Figure 4.* Absolute GRF on a log scale, $\log_{10}(\mathrm{GRF}_{\mathrm{abs}}(H))$, across temporal horizons $H$. LRMM consistently outperforms LIF, ALIF, and TC-LIF, with gaps widening as $H$ increases.

sizes. Our framework supports heterogeneous neuron integration to further enhance performance. By replacing the input encoder $N_I$ and readout neuron $N_O$ with ALIF units,

*Table 2.* Representative copy-task accuracy under a fixed delay of $\text{delay}_t = 20$. LRMM uses two layers with 128 circuits, while ALIF uses two layers with 1024 neurons. The full $(L, K)$ accuracy grid is shown in Figure 3.

| Setting | LRMM | ALIF |
|---|---|---|
| $L = 8,\ K = 8$ | 0.89 | 0.58 |
| $L = 10,\ K = 10$ | 0.67 | 0.43 |

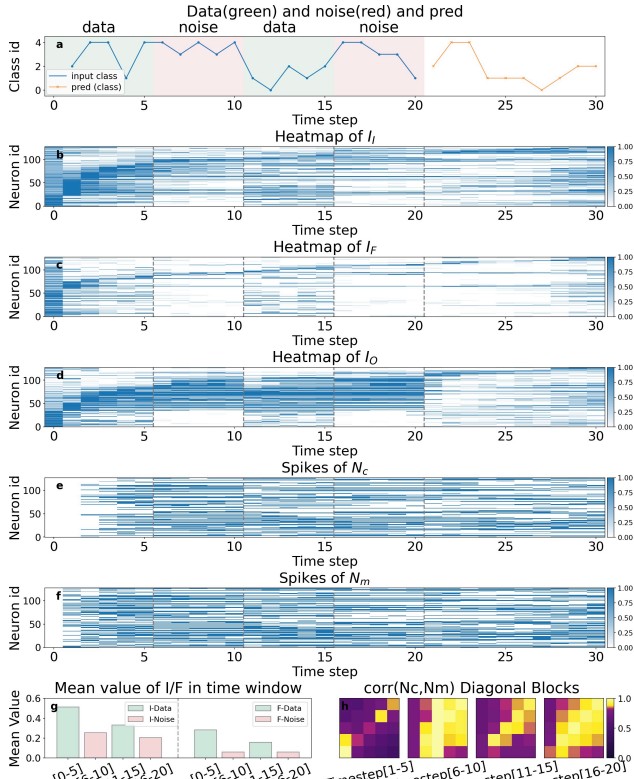

*Figure 5.* Visualization of LRMM's gating and spiking mechanisms on a 20-step input sequence. Green segments indicate informative symbols requiring memory, while red segments denote noise. LRMM exhibits selective I/F gating during informative intervals and O gating during readout, with corresponding spiking patterns and correlation structure across neuron subpopulations.

while maintaining the memory neuron $N_M$ and aggregation neuron $N_C$ as LIF, our method achieves higher accuracy of 97.39% on PS-MNIST, 80.51% on SSC, 95.32% on SHD, and 100.00% on Binary Adding under the same parameter counts.

**Evaluating Long-Term Memory.** We evaluate the long-term memory capacity of LRMM using the copy task (Graves et al., 2014; Bellec et al., 2020), a canonical benchmark for measuring temporal credit assignment across extended delays. Each input consists of a sequence of $L \in \{2, \ldots, 10\}$ tokens drawn from an alphabet of size $K \in \{2, \ldots, 10\}$, followed by a stop signal and a fixed delay of $\text{delay}_t = 20$ time steps. After receiving the readout cue, the model must reproduce the original sequence in exact order and length. Figure 3 reports the full test accuracy grid over different sequence lengths and alphabet sizes. To make the key results more readable, we summarize representative settings in Table 2. LRMM consistently outperforms a much larger ALIF baseline under challenging long-delay settings, suggesting that the memory loop between $N_M$ and $N_C$ effectively preserves task-relevant information during the delay interval.

**Adaptive Recurrent Dynamics.** To analyze how LRMM adaptively processes information across memory stages, we use a modified copy task which contains structured noise and explicit control signals, as shown in Figure 5(a). Each input sequence spans 20 time steps, structured as two cycles of alternating informative segments and noise intervals (5 steps each). At each time step, a control signal indicates whether the current input should be remembered or ignored. After the entire input sequence, a readout signal is issued, and the model must output the concatenated informative segments in order. As shown in Figures 5(b)–(d) and (g), during noise segments, both the forget signal $F$ and input signal $I$ are significantly reduced compared to informative data, indicating that the model avoids forgetting stored content while ignoring irrelevant input. At the same time, Figures 5(e), (f), and (h) show stronger Hamming correlation between $N_M$ and $N_C$, suggesting more stable internal recurrence. In contrast, during informative data, both $F$ and $I$ increase, reflecting active integration of new input with existing memory, accompanied by more dynamic activity between $N_M$ and $N_C$. During the readout phase, the output $O$ becomes

selectively active, not merely propagating memory but enabling targeted information retrieval.

These results suggest that LRMM achieves robust memory control through adaptive recurrent mechanisms that filter, store, and extract information in noisy temporal settings.

**Gradient Retention Analysis.** We further evaluate the temporal gradient stability by computing the relative Gradient Retention Factor (GRF) A.5 across training. As shown in Figure 6, LRMM achieves consistently higher single-step geometric gain compared to the baseline without inter-loop recurrence, exceeding it by more than $1.5\times$ on average. This indicates significantly improved gradient flow and more effective temporal credit assignment over long horizons. As shown in Figure 4, LRMM achieves the highest absolute GRF A.5 across all tested horizons. The gap becomes increasingly significant as $H$ grows, indicating that the structured recurrent feedback in LRMM enables more stable gradient propagation over long sequences, compared to LIF, ALIF and TC-LIF.

We further compare LRMM with EGRU and Event-SSM in

*Table 3.* Ablation study of LRMM.

| Dataset | Ablation Setting | Accuracy(%) ↑ |
|---|---|---|
| PS-MNIST | Full Model | 96.52 |
| | w/o $N_C \to N_M$ connection | $86.11_{\downarrow 9.41}$ |
| | w/o Global recurrence | $93.74_{\downarrow 2.78}$ |
| | w/o Gate Separation | $92.35_{\downarrow 4.17}$ |
| SHD | Full Model | 94.70 |
| | w/o $N_C \to N_M$ connection | $86.28_{\downarrow 8.42}$ |
| | w/o Global recurrence | $92.47_{\downarrow 2.23}$ |
| | w/o Gate Separation | $90.81_{\downarrow 3.89}$ |

*Table 4.* Energy consumption comparison on SHD dataset using two-layer networks with 512 neurons per layer.

| Model | Theoretical energy | Energy (nJ) |
|---|---|---|
| LRMM | $(7n\,E_{\text{MAC}} + n(3m\,\text{Fr}_{\text{in}} + 3n\,\text{Fr}_{N_O}/4$ $+\text{Fr}_{N_I} + \text{Fr}_{N_M} + 2\,\text{Fr}_{N_C})\,E_{\text{AC}})/4$ | 76.11 |
| TC-LIF | $2n\,E_{\text{MAC}} + (mn\,\text{Fr}_{\text{in}} + (n^2+2n)\,\text{Fr}_{\text{out}})\,E_{\text{AC}}$ | 212.27 |
| LIF | $n\,E_{\text{MAC}} + (mn\,\text{Fr}_{\text{in}} + (n^2+n)\,\text{Fr}_{\text{out}})\,E_{\text{AC}}$ | 186.60 |
| LSTM | $(4(mn+n^2) + 17n)\,E_{\text{MAC}}$ | 21145 |

Appendix A.17, where LRMM shows a favorable accuracy-efficiency trade-off against dense state-space and event-driven recurrent baselines.

## 4.3. Ablation Study

To evaluate the contribution of each structural component in our LIF memory circuit, we conduct a series of controlled ablation experiments. All variants are trained under the same protocol on PS-MNIST and SHD. We measure classification accuracy and BPTT gradient stability to assess the effect of circuit modifications. Specifically, we ablate: (1) the recurrent feedback from the context neuron $N_C$ to the memory neuron $N_M$, (2) the recurrent output path from $N_O$ to the current layer, and (3) the gating mechanism, replacing all three gates with a shared static input gate.

As shown in Table 3, all three ablations cause consistent performance drops across both datasets, validating the necessity of feedback modulation, temporal recurrence, and gate specialization in the proposed memory circuit.

**Memory-State Modulated Feedback.** Removing the feedback from the context neuron $N_C$ to the memory neuron $N_M$ results in the most significant degradation: PS-MNIST accuracy drops from 96.52% to 86.11%, and SHD drops from 94.70% to 86.28%. This ablation breaks the memory-context loop, impairing the circuit's ability to retain and coordinate long-term information.

**Recurrent Memory Path.** Eliminating the output recurrence from $N_O$ to the current layer weakens temporal integration. Accuracy drops moderately by 2.78% on PS-MNIST and 2.23% on SHD. Although the model retains basic temporal processing via internal delays, the lack of global recurrence leads to reduced gradient stability and more localized memory formation, especially in longer sequences.

**Gate Separation.** Replacing the three distinct modulatory gates $(I_I, I_F, I_O)$ with a single shared input gate impairs selective signal routing. This simplification causes accuracy to drop by 4.17% on PS-MNIST and 3.89% on SHD, suggesting that dedicated gating enables fine-grained temporal

filtering of relevant versus irrelevant information streams.

We further conduct supplementary ablations to isolate the roles of the three modulatory currents, the leak factor, and the modulation-function training scheme. As summarized in Appendix and Tables 7–9, using all three currents consistently yields the best performance, while removing any current degrades accuracy, indicating a synergistic rather than redundant contribution. We also observe that LRMM is relatively insensitive to the leak factor over a broad range, and that the modulation-function training improves optimization stability and accuracy compared to removing modulation.

## 4.4. High Energy Efficiency

We estimate dynamic energy using the standard operation-counting model:

$$E = \#\text{MAC} \cdot E_{\text{MAC}} + \#\text{AC} \cdot E_{\text{AC}}, \qquad (22)$$

where $E_{\text{AC}} = 0.9$ pJ and $E_{\text{MAC}} = 4.6$ pJ following Horowitz (2014). We evaluate energy consumption on SHD using two hidden layers with 512 neurons per layer. The detailed firing rates used in the estimation are in Appendix A.6.

**Energy efficiency.** On SHD, LRMM attains the lowest dynamic energy among all recurrent baselines, as shown in Table 4. Thanks to its localized memory loops and sparsely activated recurrent connections, spike activity is concentrated within a small subset of synapses, which substantially reduces the number of event-driven accumulation operations compared to more densely activated LIF/TC-LIF layers and gate-based LSTMs. Based on the operation-counting estimation, LRMM consumes an estimated 76.11 nJ per inference step, compared with 186.60 nJ for LIF and 212.27 nJ for TC-LIF, corresponding to approximately 2.45× and 2.79× lower estimated dynamic energy, respectively. LSTM is even more expensive, consuming 21145 nJ per step, so LRMM is roughly 277× more energy-efficient. Details are in Appendix A.8. A fine-grained, operation-level energy breakdown is reported in Appendix A.13 and Table 10. Overall, LRMM trades a small linear increase in MACs for a much larger reduction in spike-driven accumulation events, yielding the best energy–performance balance among all compared models on SHD. We further analyze spike communication, memory access, and training-time overhead in

Appendix A.18.

## 5. Summary and Discussion

We introduced the LIF Recurrent Memory Module (LRMM), a lightweight spiking memory architecture built entirely from vanilla LIF neurons with fixed, structured connectivity. By augmenting LIF dynamics with a localized memory loop, LRMM achieves long-range temporal integration while preserving low firing rates, low parameter count, and high energy efficiency. The architecture promotes stable gradient flow, enabling effective temporal credit assignment without relying on adaptive threshold mechanisms or explicit synaptic delays. Extensive experiments on benchmark sequence tasks demonstrate that LRMM achieves high performance among SNNs, while consuming up to $59\%$ less event-driven energy than LIF and over $277\times$ less energy than LSTM.

**Limitations and Future Work.** Despite these advantages, LRMM still has several limitations. First, our experiments mainly focus on medium-scale long-sequence benchmarks, while its scalability to ultra-long tasks such as Long Range Arena, language modeling, long-context reasoning, and decision making remains to be explored. Second, the reported energy efficiency is based on an operation-counting estimation rather than direct neuromorphic hardware deployment, where limited precision, routing overhead, and core-mapping constraints may further affect practical efficiency. Third, LRMM is currently evaluated as a recurrent spiking architecture, and extending it to larger hierarchical memory systems or hybrid architectures with Transformers and state-space models is an important future direction.

## Acknowledgments

This work was supported in part by the National Key R&D Plan of China (2023YFF1203600), Natural Science Foundation of China (62574180), Pioneer R&D Program of Zhejiang (2026C01008) and the Open Fund of the State Key Laboratory of Integrated Optoelectronics.

## Impact Statement

This paper presents work whose goal is to advance the field of Machine Learning. There are many potential societal consequences of our work, none which we feel must be specifically highlighted here.

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

# A. Appendix

### A.1. BPTT proof

**Notation.** We index time $t$ and units $X \in \{N_I, N_M, N_C, N_O\}$ and current gates $j \in \{I, F, O\}$. Each unit keeps membrane $U_X[t]$, spike $S_X[t]$, input $I_X[t]$, leak $\alpha_X \in (0, 1)$, threshold $V_{\text{th}}$. We reuse $gZ[t] = \partial\mathcal{L}/\partial Z[t]$ for any symbol $Z$ (e.g., $gU_X[t], gS_X[t], gI_j[t]$) and parameter gradients $\partial\mathcal{L}/\partial w$.

**Soft-reset update and derivation of $A_X[t]$.** During training we adopt a soft-reset LIF update

$$U_X[t{+}1] \;=\; \alpha_X(U_X[t] - V_{\text{th}}S_X[t]) + I_X[t{+}1], \tag{23}$$

and treat $I_X[t{+}1]$ as exogenous when differentiating w.r.t. $U_X[t]$ so that inter-step structural effects are accounted for separately in the spatial graph. With the surrogate $S_X[t] = \sigma_X(U_X[t] - V_{\text{th}})$ and $\partial S_X[t]/\partial U_X[t] = \sigma'_X(\cdot)$, we obtain

$$A_X[t] \;=\; \frac{\partial U_X[t{+}1]}{\partial U_X[t]} \;=\; \alpha_X\Big(1 - V_{\text{th}}\,\sigma'_X(U_X[t] - V_{\text{th}})\Big). \tag{24}$$

This coefficient quantifies the local temporal gain induced jointly by membrane leak and subtract-threshold reset.

**Surrogate gradients (training only).** We keep the forward dynamics hard and invoke surrogates only in the backward pass. For the modulation clamp in (4), when the forward path uses the piecewise-linear hard-sigmoid $\text{PL}(z)$, its derivative is approximated by the logistic-sigmoid derivative:

$$\frac{\partial I_j[t]}{\partial z_j[t]} \;\approx\; \sigma(z_j[t])(1 - \sigma(z_j[t])), \qquad z_j[t] \;=\; W_j\,[\,\text{input}[t];\, S_O[t{-}1]\,] + b_j, \tag{25}$$

where $\sigma(z) = 1/(1 + e^{-z})$. For spikes in (3), we adopt the rectangular surrogate with half-width $H_w > 0$:

$$\frac{\partial S_X[t]}{\partial U_X[t]} \;=\; \begin{cases} \dfrac{1}{2H_w}, & |U_X[t] - V_{\text{th}}| \leq H_w, \\ 0, & \text{otherwise.} \end{cases} \tag{26}$$

Gradients through the reset factor $(1 - S_X[t])$ use the same spike surrogate $\partial S_X[t]/\partial U_X[t]$. At inference time we employ the hard $\text{PL}(\cdot)$ clamp and the Heaviside firing function without surrogates.

**Step-by-step BPTT for each neuron in details.** We write $\sigma'_X[t] \equiv \sigma'_X(U_X[t] - V_{\text{th}})$ and use $A_X[t]$ from (11). Input neuron $N_I$:

$$gU_I[t] = \underbrace{A_I[t]\,gU_I[t+1]}_{\text{temporal}} + \sigma'_I[t]\underbrace{w_{I,C}\,gU_C[t]}_{\text{spatial}} \tag{27}$$

Neuron $N_M$:

$$gU_M[t] = \underbrace{A_M[t]\,gU_M[t+1]}_{\text{temporal}} + \sigma'_M[t]\underbrace{w_{M,C}\,gU_C[t]}_{\text{spatial}} \tag{28}$$

Neuron $N_C$:

$$gU_C[t] = \underbrace{A_C[t]\,gU_C[t+1]}_{\text{temporal}} + \sigma'_C[t](\underbrace{w_{C,O}\,gU_O[t]}_{\text{spatial}} + \underbrace{w_{C,M}\,gU_M[t+1]}_{\text{temporal}}) \tag{29}$$

Neuron $N_O$:

$$gU_O[t] = \underbrace{\Big(A_O[t] + \sigma'_O[t]\,c_O[t+1]w_{N_O,O}k_O\Big)gU_O[t+1]}_{\text{temporal}}$$

$$+ \sigma'_O[t]\Big(\underbrace{c_I[t+1]w_{N_O,I}k_I\,gU_I[t+1]}_{\text{temporal}} + \underbrace{c_F[t+1]w_{N_O,F}k_F\,gU_M[t+1]}_{\text{temporal}} + \underbrace{\frac{\partial\mathcal{L}}{\partial S_O[t]}}_{\text{spatial}}\Big) \tag{30}$$

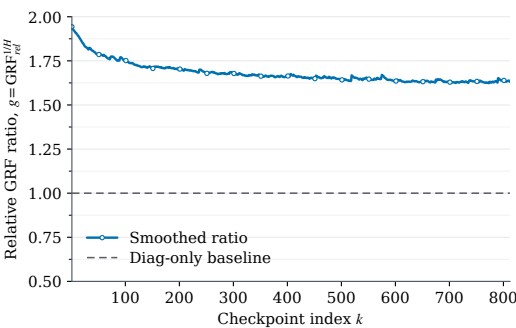

*Figure 6.* Single-step geometric gain $g = (\mathrm{GRF}_{\mathrm{rel}})^{1/H}$ on SHD ($H = 32$). LRMM shows consistently higher gain than the baseline without inter-loop connections.

Here $c_j[t+1]$ collects the local slope along the path $S_O[t] \rightarrow I_j[t+1]$ through the modulation $\Phi$ and the corresponding linear map, for $j \in \{\mathrm{I}, \mathrm{F}, \mathrm{O}\}$.

### A.2. Datasets

**SHD (Spiking Heidelberg Digits)** (Cramer et al., 2020) is a neuromorphic dataset that consists of spike-based representations of spoken digits (0–9), recorded using a model of the auditory periphery. Each sample is represented as a sequence of spatio-temporal spike events across 700 input channels over a duration of 1 second. It is particularly suited for evaluating the temporal processing capabilities of spiking neural networks (SNNs).

**SSC (Spiking Speech Commands)** (Warden, 2018) is a spike-based speech-command recognition benchmark derived from the Google Speech Commands dataset. It converts audio samples into spatio-temporal spike trains using biologically inspired auditory models, and evaluates models on speech-command classification under the benchmark-defined label protocol. Like SHD, SSC emphasizes precise temporal integration and robustness in spike-based auditory representations, making it a suitable testbed for SNN-based models.

**PSMNIST (Permuted Sequential MNIST)** (Le et al., 2015) is a sequential version of the standard MNIST handwritten digit dataset. Each 28×28 image is flattened into a 784-dimensional sequence, and then a fixed random permutation is applied to the sequence order. The dataset contains 60,000 training and 10,000 test samples. PSMNIST is widely used to benchmark recurrent and sequential models due to its requirement for long-range dependency modeling.

**Binary Adding (long-range marked-sum).** Following (Ma et al., 2025), this synthetic sequence task is designed to evaluate a model's ability to capture long-range temporal dependencies. Each input contains two binary sequences of length $T$: a value sequence $x_1 \in \{0, 1\}^T$ and a marker sequence $x_2 \in \{0, 1\}^T$. The marker $x_2$ selects 9 positions within $x_1$, and the label is the sum of $x_1$ at these positions, yielding a 10-class target (0–9). The model must process the entire sequence before prediction, making it a strict test of temporal integration. We generate 50,000 training and 2,000 test samples, and vary $T$ to control task difficulty.

### A.3. Training Details

All LRMM units share the same LIF parameters: a trainable leak factor initialized to 0.95, a fixed firing threshold $V_{\mathrm{th}} = 1.0$, and a reset potential $V_{\mathrm{reset}} = 0$. All feedforward and recurrent weights are initialized using Xavier uniform initialization. We adopt a boxcar surrogate gradient with width $w = 1.0$. Full input sequences are used without truncation during BPTT. Training is performed using the Adam optimizer with $\beta_1 = 0.9$ and $\beta_2 = 0.999$, a batch size of 128, a total of 100 epochs and an initial learning rate of $1 \times 10^{-3}$ for PS-MNIST and $1 \times 10^{-2}$ for other datasets. Classification is performed based on the firing rates of the output neurons, and the model is trained using the standard cross-entropy loss.

### A.4. Baseline Models and Comparative Methodology

We evaluate three categories of baselines under controlled model capacity and training settings. First, to test whether network-level structure can replace neuron-level complexity, we compare against ALIF (Yin et al., 2021), TC-LIF (Zhang et al., 2024a) and RadLIF (Bittar & Garner, 2022), which incorporate adaptive thresholds, compartmental dynamics or

*Table 5.* Firing rates used for the SHD energy estimation. For LRMM, firing rates are reported for different neuron groups in the local recurrent memory module.

| Model | Internal firing rates | Output firing rate |
|---|---|---|
| LRMM | $Fr_{N_I} = (0.168, 0.196)$, $Fr_{N_O} = (0.311, 0.269)$, $Fr_{N_C} = (0.366, 0.330)$, $Fr_{N_M} = (0.274, 0.197)$ | $Fr_{\text{out}} = 0.080$ |
| LIF | $(0.274, 0.226)$ | $Fr_{\text{out}} = 0.085$ |
| TC-LIF | $(0.294, 0.241)$ | $Fr_{\text{out}} = 0.108$ |

radial memory. Second, we include an LSTM (Hochreiter & Schmidhuber, 1997) with matched parameter budget to assess compute cost and energy efficiency relative to a standard ANN baseline. Third, we test a stacked LIF network without the LRMM loop, isolating the effect of our proposed structural design. These comparisons help disentangle neuron-intrinsic mechanisms from architectural complexity, and quantify the impact of LRMM under consistent experimental conditions.

Regarding training configurations, ALIF, RadLIF, TC-LIF use normalization layers. For surrogate gradients, ALIF uses a triangular surrogate, RadLIF uses a Gaussian surrogate, TC-LIF uses a triangular surrogate.

## A.5. Metrics for Temporal Stability and Gradient Propagation

We use a compact metric suite to characterize (i) global multi-step gradient retention and (ii) the local stability of the LRMM microcircuit.

**Absolute Gradient Retention Factor (GRF).** Let $J_t = \partial u_{t+1}/\partial u_t$ be the per-step Jacobian of a neuron/microcircuit state. Over horizon $H$ starting at $t$,

$$\text{GRF}_{\text{abs}}(H; t) = \Big\| \prod_{k=0}^{H-1} J_{t+k} \Big\|_2, \quad \text{GRF}_{\text{abs}}(H) = \text{median}_{t \in \mathcal{T}} \, \text{GRF}_{\text{abs}}(H; t).$$

Larger values indicate stronger long-range gradient preservation.

**Loop-Sensitive Relative GRF (LRMM).** To isolate the $N_M \leftrightarrow N_C$ loop effect, define the LRMM temporal operator

$$M_t = \begin{bmatrix} A_M^t + \gamma_t \beta_t & \gamma_t A_C^t \\ \beta_t & A_C^t \end{bmatrix}, \quad D_t = \text{diag}(A_M^t, A_C^t),$$

with $A_X^t = \alpha_X(1 - V_{\text{th}} \, \sigma'_X(U_X^t - V_{\text{th}}))$, $\beta_t = \sigma'_C(U_C^t - V_{\text{th},N_C}) \, w_{C,M}$, $\gamma_t = \sigma'_M(U_M^t - V_{\text{th},N_M}) \, w_{M,C}$, where $\sigma'$ is the surrogate derivative. The loop contribution is

$$\text{GRF}_{\text{rel}}(H; t) = \frac{\| \prod_{k=0}^{H-1} M_{t+k} \|_2}{\| \prod_{k=0}^{H-1} D_{t+k} \|_2}, \quad \text{GRF}_{\text{rel}}(H) = \text{median}_{t \in \mathcal{T}} \, \text{GRF}_{\text{rel}}(H; t).$$

Values $> 1$ indicate loop-induced amplification beyond diagonal decay.

**Spectral Radius.** Local stability is summarized by $\rho(M_t) = \max_i |\lambda_i(M_t)|$: $\rho(M_t) < 1$ suggests contraction, $\rho(M_t) \approx 1$ near-critical memory, and $\rho(M_t) > 1$ potential instability.

## A.6. Firing Rates Used for Energy Estimation

Table 5 reports the firing-rate statistics used in the energy estimation of Table 4. These statistics are measured on the SHD benchmark under the same two-layer network setting used in Sec. 4.4. For LRMM, we report the firing rates of different neuron groups within the local recurrent memory module, including $N_I$, $N_M$, $N_C$, and $N_O$. For LIF and TC-LIF, we report the layer-wise firing rates and output firing rates used in the corresponding energy formulas.

## A.7. Energy Counting of LRMM under the Event-Driven Convention

**Counting convention.** We follow the *event-driven* convention used in prior SNN energy tables: (i) event-triggered computations (including projections, gates, and loop updates) are accounted as accumulations with unit cost $E_{\text{AC}}$; (ii) only the per-neuron temporal state update (leakage/reset) is treated as a dense multiply–accumulate (MAC), yielding $n E_{\text{MAC}}$ per step.

**Notation.** Let $m$ be the input width and $n$ the total number of neurons in the layer. Denote by $Fr_{\text{in}}$ the average input firing ratio, and by $Fr_{N_O}$ the average output firing ratio (neuron $N_O$). Within the local loop, $Fr_{N_I}$, $Fr_{N_M}$, and $Fr_{N_C}$ denote the firing ratios associated with nodes $N_I$, $N_M$, and $N_C$, respectively. The symbols $E_{\text{MAC}}$ and $E_{\text{AC}}$ denote the unit energy of a MAC and an accumulation, respectively. Event-triggered operations are charged as $E_{\text{AC}}$; temporal leakage/reset contributes the $nE_{\text{MAC}}$ baseline.

**Operation counting per step.** We decompose the per-step energy into four parts and then aggregate.

**(A) Per-neuron temporal state update.** Each neuron incurs one dense update per step,

$$E_A = n\, E_{\text{MAC}}. \tag{31}$$

**(B) Input-driven accumulations.** The three gates consume input events over $m \times \frac{n}{4}$ connections,

$$E_B = \frac{3}{4} mn\, Fr_{\text{in}}\, E_{\text{AC}}. \tag{32}$$

**(C) Output-driven accumulations.** Output spikes at node $N_O$ fan out to all $n/4$ local modules along three paths per module,

$$E_C = \frac{3}{16} n^2\, Fr_{N_O}\, E_{\text{AC}}. \tag{33}$$

**(D) Two-node loop accumulations.** Within each module, the loop $(N_M \leftrightarrow N_C)$ and other connections add local event interactions. Aggregating over all modules yields,

$$E_D = \frac{1}{4} n(Fr_{N_I} + Fr_{N_M} + 2\, Fr_{N_C})\, E_{\text{AC}}. \tag{34}$$

**(E) Input current scaling.** The three input currents $I_I$, $I_F$, and $I_O$ each receive a learned scalar weight,

$$E_E = \frac{3}{4} n\, E_{\text{MAC}}. \tag{35}$$

**Aggregate cost.** Summing (31)–(35) gives

$$E_{\text{LRMM/step}} = \frac{7}{4} n\, E_{\text{MAC}} + \left( \frac{3mn}{4} Fr_{\text{in}} + \frac{3n^2}{16} Fr_{N_O} + \frac{n}{4}(Fr_{N_I} + Fr_{N_M} + 2Fr_{N_C}) \right) E_{\text{AC}}. \tag{36}$$

### A.8. Energy Counting for TC-LIF, LIF, LSTM on SHD

**Setup.** Two hidden layers with total neurons per layer $n = 512$. Layer-1 takes external input of width $m_1 = 700$. Layer-2 takes the output of Layer-1 with effective width $m_2 = 512$. The theoretical per-step energies used are

$$E_{\text{TC-LIF}}^{(\ell)} = 2n\, E_{\text{MAC}} + (m_\ell n\, Fr_{\text{in}}^{(\ell)} + (n^2 + 2n)\, Fr_{\text{out}})\, E_{\text{AC}}, \tag{37}$$

$$E_{\text{LIF}}^{(\ell)} = n\, E_{\text{MAC}} + (m_\ell n\, Fr_{\text{in}}^{(\ell)} + (n^2 + n)\, Fr_{\text{out}})\, E_{\text{AC}}, \tag{38}$$

$$E_{\text{LSTM}}^{(\ell)} = (4(m_\ell n + n^2) + 17n)\, E_{\text{MAC}}. \tag{39}$$

**Firing rates.** LIF: $(Fr_{\text{in}}^{(1)}, Fr_{\text{in}}^{(2)}) = (0.274,\ 0.226)$, $Fr_{\text{out}} = 0.085$.
TC-LIF: $(Fr_{\text{in}}^{(1)}, Fr_{\text{in}}^{(2)}) = (0.294,\ 0.241)$, $Fr_{\text{out}} = 0.108$.

**LIF**

Layer-1:

$$E_{\text{LIF}}^{(1)} = 512\, E_{\text{MAC}} + (700 \cdot 512 \cdot 0.274 + (512^2 + 512) \cdot 0.085)\, E_{\text{AC}}$$
$$= 512\, E_{\text{MAC}} + 120527.36\, E_{\text{AC}}. \tag{40}$$

Layer-2:

$$E_{\text{LIF}}^{(2)} = 512\,E_{\text{MAC}} + \left(512 \cdot 512 \cdot 0.226 + (512^2 + 512) \cdot 0.085\right) E_{\text{AC}}$$
$$= 512\,E_{\text{MAC}} + 81570.304\,E_{\text{AC}}. \tag{41}$$

Two-layer total:

$$E_{\text{LIF,total}} = 1024\,E_{\text{MAC}} + 202097.664\,E_{\text{AC}} = 186.60\text{nJ}. \tag{42}$$

**TC-LIF**

Layer-1:

$$E_{\text{TC-LIF}}^{(1)} = 1024\,E_{\text{MAC}} + \left(700 \cdot 512 \cdot 0.294 + (512^2 + 2 \cdot 512) \cdot 0.108\right) E_{\text{AC}}$$
$$= 1024\,E_{\text{MAC}} + 133791.744\,E_{\text{AC}}. \tag{43}$$

Layer-2:

$$E_{\text{TC-LIF}}^{(2)} = 1024\,E_{\text{MAC}} + \left(512 \cdot 512 \cdot 0.241 + (512^2 + 2 \cdot 512) \cdot 0.108\right) E_{\text{AC}}$$
$$= 1024\,E_{\text{MAC}} + 91598.848\,E_{\text{AC}}. \tag{44}$$

Two-layer total:

$$E_{\text{TC-LIF,total}} = 2048\,E_{\text{MAC}} + 225390.592\,E_{\text{AC}} = 212.27\text{nJ}. \tag{45}$$

**LSTM**

Layer-1:

$$E_{\text{LSTM}}^{(1)} = \left(4(700 \cdot 512 + 512^2) + 17 \cdot 512\right) E_{\text{MAC}}$$
$$= 2490880\,E_{\text{MAC}}. \tag{46}$$

Layer-2:

$$E_{\text{LSTM}}^{(2)} = \left(4(512 \cdot 512 + 512^2) + 17 \cdot 512\right) E_{\text{MAC}}$$
$$= 2105856\,E_{\text{MAC}}. \tag{47}$$

Two-layer total:

$$E_{\text{LSTM,total}} = 4596736\,E_{\text{MAC}} = 21145\text{nJ}. \tag{48}$$

### A.9. Energy Counting of LRMM on SHD

**Setup.** Two LRMM layers, each with $n_h = 512$ neurons split evenly: $|N_I| = |N_M| = |N_C| = |N_O| = 128$. Layer–1 uses external input of width $m_1 = 700$ with firing rate $Fr_{\text{in}}^{(1)} = 0.114$. Layer–2 takes input from Layer–1's output sub-population, hence $m_2 = |N_O^{(1)}| = 128$ and $Fr_{\text{in}}^{(2)} = Fr_{N_O}^{(1)}$. A final linear readout maps $|N_O^{(2)}| = 128$ to 20 classes. Per-neuron temporal updates contribute $n_h E_{\text{MAC}}$ per step, and all event-triggered operations (input projections, output fan-out, and local-module interactions) are accounted as accumulation events with cost $E_{\text{AC}}$.

**Per-layer counting (using total $n$).** Let $n = 512$ be the total number of hidden neurons per LRMM layer. For layer $\ell \in \{1, 2\}$ with input width $m_\ell$ and input firing $Fr_{\text{in}}^{(\ell)}$, the per-step energy is

$$E_{\text{step}}^{(\ell)} = E_A^{(\ell)} + E_B^{(\ell)} + E_C^{(\ell)} + E_D^{(\ell)} + E_E^{(\ell)}$$
$$= \frac{7}{4} n\,E_{\text{MAC}} + \left(\frac{3m_\ell n}{4} Fr_{\text{in}}^{(\ell)} + \frac{3n^2}{16} Fr_{N_O}^{(\ell)} + \frac{n}{4}\left(Fr_{N_I}^{(\ell)} + Fr_{N_M}^{(\ell)} + 2\,Fr_{N_C}^{(\ell)}\right)\right) E_{\text{AC}}. \tag{49}$$

*Table 6.* Total per-step energy on SHD (two hidden layers, $n$=512 each). Counts follow $E_{\text{total}} = \#\text{MAC} \cdot E_{\text{MAC}} + \#\text{AC} \cdot E_{\text{AC}}$ with $E_{\text{MAC}} = 4.6\,\text{pJ}$ and $E_{\text{AC}} = 0.9\,\text{pJ}$. LRMM total includes the $128 \to 20$ readout.

| Model (2 layers) | Theoretical per-step count | Measured energy (nJ) |
|---|---|---|
| LRMM (2 layers + out) | $1792\,E_{\text{MAC}} + 75{,}411.328\,E_{\text{AC}}$ | **76.11** |
| LIF (2 layers) | $1024\,E_{\text{MAC}} + 202{,}097.664\,E_{\text{AC}}$ | 186.5983 |
| TC-LIF (2 layers) | $2048\,E_{\text{MAC}} + 225{,}390.592\,E_{\text{AC}}$ | 212.2723 |
| LSTM (2 layers) | $4{,}596{,}736\,E_{\text{MAC}} + 0\,E_{\text{AC}}$ | 21,144.9856 |

**Firing rates (given).**

$$\text{Layer 1: } (Fr^{(1)}_{N_I}, Fr^{(1)}_{N_O}, Fr^{(1)}_{N_C}, Fr^{(1)}_{N_M}) = (0.168,\ 0.311,\ 0.366,\ 0.274),$$

$$\text{Layer 2: } (Fr^{(2)}_{N_I}, Fr^{(2)}_{N_O}, Fr^{(2)}_{N_C}, Fr^{(2)}_{N_M}) = (0.196,\ 0.269,\ 0.330,\ 0.197).$$

**Layer–1 ($m_1$=700, $Fr^{(1)}_{\text{in}}$=0.114).** Substituting into (49) gives

$$E_{\text{L1}} = 896\,E_{\text{MAC}} + 46079.744\,E_{\text{AC}}.$$

**Layer–2 ($m_2$=128, $Fr^{(2)}_{\text{in}}=Fr^{(1)}_{N_O}$=0.311).**

$$E_{\text{L2}} = 896\,E_{\text{MAC}} + 28642.944\,E_{\text{AC}}.$$

**Final readout ($|N^{(2)}_O|$=128 $\to$ 20).** Under the same event-driven assumption, the linear readout cost per step is

$$E_{\text{readout}} = (128 \times 20 \times Fr^{(2)}_{N_O})\,E_{\text{AC}} = (2560 \times 0.269)\,E_{\text{AC}} = 688.64\,E_{\text{AC}}. \tag{50}$$

**Two-layer total per step (with readout).** Summing the two LRMM layers and the readout yields

$$E_{\text{total/step}} = 1792\,E_{\text{MAC}} + 75411.328\,E_{\text{AC}}. \tag{51}$$

## A.10. Ablation Study on Current Combinations.

*Table 7.* Ablation of current combinations with test accuracy on PS-MNIST.

| Currents Combo | $I_I+I_F+I_O$ | $I_I+I_F$ | $I_I+I_O$ | $I_F+I_O$ | $I_I$ | $I_F$ | $I_O$ |
|---|---|---|---|---|---|---|---|
| Acc | 96.52 | 94.45 | 93.14 | 93.60 | 90.28 | 89.76 | 84.91 |

Table 7 investigates how different combinations of currents affect performance on PS-MNIST. Using the full triplet ($I_I+I_F+I_O$) yields the highest accuracy of 96.52%, while removing any single current consistently degrades performance: the best two-current variant $I_I+I_F$ drops to 94.45%, and $I_I+I_O$ and $I_F+I_O$ further decrease to 93.14% and 93.60%, respectively. When only a single current is retained, accuracy falls to around 90% for $I_I$ and $I_F$, and down to 84.91% for $I_O$ alone. These results indicate that the three currents contribute synergistically rather than redundantly, and that the input and forget currents ($I_I$ and $I_F$) are particularly crucial for maintaining high task performance, while $I_O$ alone is insufficient but still provides complementary gains when combined with the others.

## A.11. Ablation Experiments on Initialization of Leak Factor

For parameter sensitivity, we conduct a grid search by varying the initialization of the leak factor and measuring the resulting test accuracy on PS-MNIST. The leak factor is defined as $LF = 1 - 1/\tau$, i.e., the per-timestep decay multiplier applied to the membrane potential.

As shown in Table 8, LRMM remains stable across a wide range of initial leak factors. Even with strong leakage ($LF = 0.10$) or slow decay ($LF = 0.99$), the model consistently achieves around 95–96.5% accuracy, with the best performance at $LF = 0.95$. This weak sensitivity is expected because the leak parameters are learnable in LRMM: during training, the model can adapt the effective leak to a suitable regime, so the initialization mainly affects convergence speed rather than the final accuracy.

*Table 8.* Parameter Sensitivity to the Leak Factor $LF = 1 - 1/\tau$ Measured on PS-MNIST.

| $LF$ | 0.10 | 0.20 | 0.30 | 0.40 | 0.50 | 0.60 | 0.70 | 0.80 | 0.90 | 0.95 | 0.99 |
|------|------|------|------|------|------|------|------|------|------|------|------|
| Acc | 95.11 | 95.07 | 95.63 | 95.48 | 95.87 | 96.08 | 96.17 | 96.04 | 96.36 | 96.52 | 96.49 |

## A.12. Ablation Study on Modulation Function Training

*Table 9.* Ablation study on the modulation function: test accuracy (%) on PS-MNIST and SHD with and without the modulation mechanism.

| Benchmarks | PS-MNIST | SHD |
|------------|----------|------|
| w/ modulation | 96.52 | 94.70 |
| w/o modulation | 92.81 | 93.17 |

Table 9 evaluates the impact of the modulation function on PS-MNIST and SHD. With the modulation mechanism enabled, LRMM achieves 96.52% and 94.70% test accuracy on PS-MNIST and SHD, respectively. Removing the modulation function leads to a clear performance drop on both benchmarks, down to 92.81% on PS-MNIST and 93.17% on SHD.

The modulation function prevents the sigmoidal gating term from saturating by dynamically linearizing its effective range during training. Without this mechanism, replacing the gating nonlinearity with a static linear or clipped function makes the recurrent dynamics more likely to get stuck in poor local optima, resulting in less stable optimization and degraded accuracy. This ablation therefore confirms that the modulation function is an important component for learning reliable gating behaviour and approaching the best performance of LRMM on temporal benchmarks.

## A.13. Fine-Grained Operation-Level Energy Analysis

*Table 10.* Theoretical energy breakdown per operation type, grouped by their corresponding equations. MAC denotes multiply–accumulate, AC denotes accumulation of spike-driven events.

| Related Eq(s) | Op. Type | Description | Energy (Table 4, nJ) |
|---------------|----------|-------------|----------------------|
| Eq. 3 | MAC | per-neuron deterministic update | 4.71 |
| Eq. 4 | AC | spike-driven forward connections | 41.95 |
| Eq. 4 | AC | spike-driven recurrent connections | 25.66 |
| Eq. 7, current terms in 8 10 | MAC | current scaling | 3.53 |
| Eq. 9, event terms in 8 10 | AC | local LRMM module interactions | 0.26 |
| Total | | | 76.11 |

To better understand where energy is spent in LRMM, we break down its per-step consumption by operation type, as shown in Table 10. Most of the energy comes from spike-driven accumulations, especially from forward and recurrent connections, which together make up over 88% of the total. In comparison, the deterministic updates and the modulation-related computations, such as current scaling and local interactions, contribute much less. This shows that while LRMM introduces additional structure for memory, it does so with only a small increase in energy, making it a practical choice for efficient neuromorphic computing.

## A.14. Analysis of Training Time and GPU Memory Usage

Table 11 compares the per-epoch training time, GPU memory usage, and accuracy of LRMM with LIF-RSNN and TC-LIF across four benchmarks. All models use two layers and 512 neurons per layer. LRMM requires more training time than the standard LIF-RSNN because the local recurrent memory module introduces additional computations.

Moreover, compared with TC-LIF, a representative neuron-centric model for long-sequence tasks, LRMM is consistently faster and uses less GPU memory across all four benchmarks. These results suggest that although LRMM introduces moderate training overhead compared with a vanilla LIF-RSNN, it achieves a more favorable trade-off among accuracy, training cost, and memory usage than more complex neuron-level alternatives.

*Table 11.* Comparison of per-epoch training time, GPU memory usage, and accuracy. All models use two layers and 512 neurons per layer.

| Benchmark | LRMM | LIF-RSNN | TC-LIF |
|---|---|---|---|
| *Training Time per Epoch* | | | |
| SHD | 41s | 16s | 51s |
| PS-MNIST | 46m39s | 13m08s | 58m27s |
| SSC | 8m32s | 3m20s | 8m58s |
| Binary Adding Problem | 35s | 12s | 39s |
| *GPU Memory Usage (MB)* | | | |
| SHD | 754 | 536 | 1250 |
| PS-MNIST | 3144 | 1374 | 4480 |
| SSC | 750 | 532 | 1246 |
| Binary Adding Problem | 714 | 508 | 1182 |

---

**Algorithm 1** One-layer LRMM forward computation

---

1: **Input:** $\{\text{input}[t]\}_{t=1}^{T}$
2: **Output:** $\{S_O[t]\}_{t=1}^{T}$
3: Initialize $U_X[0] = 0$, $S_X[0] = 0$ for $X \in \{N_I, N_M, N_C, N_O\}$
4: **for** $t = 1$ to $T$ **do**
5:     $\text{concat}[t] = [\text{input}[t]; S_O[t-1]]$
6:     $I_I[t] = \Phi(W_I \cdot \text{concat}[t] + b_I)$
7:     $I_F[t] = \Phi(W_F \cdot \text{concat}[t] + b_F)$
8:     $I_O[t] = \Phi(W_O \cdot \text{concat}[t] + b_O)$
9:     $I_{N_I}[t] = k_I \, I_I[t]$
10:    $I_{N_M}[t] = w_{C,M} S_C[t-1] + k_F I_F[t]$
11:    $I_{N_C}[t] = w_{I,C} S_I[t] + w_{M,C} S_M[t]$
12:    $I_{N_O}[t] = w_{C,O} S_C[t] + k_O I_O[t]$
13:    **for all** $X \in \{N_I, N_M, N_C, N_O\}$ **do**
14:       $U_X[t] = \alpha_X(U_X[t-1] - V_{\text{th}}S_X[t-1]) + I_X[t]$
15:       $S_X[t] = \mathbf{1}(U_X[t] > V_{\text{th}})$
16:    **end for**
17: **end for**

---

## A.15. Pseudocode for LRMM Forward Computation

This pseudocode summarizes the core structure of LRMM's forward pass over time. It highlights the modular interaction between input, memory, context, and output subpopulations, as well as the current computation and event-driven LIF updates. While conceptually simple, the local recurrent loops enable rich temporal integration with minimal global recurrence.

## A.16. Evaluation Protocol

We use Top-1 accuracy as the evaluation metric for all classification benchmarks. For all datasets, we follow the official train, validation and test splits whenever available. For datasets with a validation split, the validation set is used only for training-time model selection. For datasets without a validation split, we train the model without validation and evaluate the test set only after training is completed.

All baseline comparisons in the main paper follow the reported settings from prior work when available. For our LRMM variants, we use the same data splits, preprocessing, and evaluation metric across all compared models.

## A.17. Comparisons with Event-Driven and State-Space Baselines

We further compare LRMM with representative event-driven and state-space sequence models, including EGRU and Event-SSM. As shown in Table 12, LRMM consistently outperforms EGRU on SSC, PS-MNIST, and SHD. Compared with Event-SSM, LRMM achieves comparable accuracy on PS-MNIST and SHD, while using sparse spiking computation based on vanilla LIF neurons. Although Event-SSM achieves higher accuracy on SSC, it relies on dense state-space computation,

*Table 12.* Additional comparison with event-driven and state-space sequence baselines. Energy is estimated per step per sample on SHD under the same operation-counting energy model.

| Method | LRMM | Event-SSM | EGRU |
|---|---|---|---|
| Spiking model | Yes | No | Yes |
| SSC Acc. (%) | 79.75 | 88.4 | 71.06 |
| Params on SSC | 0.20M | 0.6M | 2.31M |
| PS-MNIST Acc. (%) | 96.52 | 97.17 | 95.1 |
| Params on PS-MNIST | 0.15M | 0.38M | 1M |
| SHD Acc. (%) | 94.70 | 95.9 | 90.47 |
| Params on SHD | 0.42M | 0.4M | 2.30M |
| Energy on SHD | 76.11 nJ | 44,788 nJ | 2,514 nJ |

*Table 13.* Spike communication statistics on SHD. All models use the same network size with 1044 neurons.

| Metric | LRMM | LIF | TC-LIF |
|---|---|---|---|
| Spikes per time step | 271.81 | 257.70 | 276.08 |
| Local spikes per time step | 195.96 | 0 | 0 |
| Spikes per sequence | 27,181 | 25,770 | 27,608 |

whereas LRMM provides a more favorable accuracy-efficiency trade-off for spiking and neuromorphic settings.

### A.18. Spike Communication Cost

We analyze the spike communication cost of LRMM on SHD using the same network size as the energy analysis, with 1044 neurons in total. As shown in Table 13, LRMM has a similar number of total spikes per time step and per sequence compared with LIF and TC-LIF. However, a large portion of LRMM spikes are local spikes within the LRMM cell. Specifically, about 72% of the spikes occur among $N_I$, $N_C$, and $N_M$, which are local to the LRMM module and do not require core-to-core communication. This hierarchical local-global recurrent design can reduce communication overhead compared with recurrent models that rely mainly on global recurrent connections.

### A.19. Scope and Stability Boundary of the Loop-Induced Gain

The $N_M$–$N_C$ loop provides an additional pass-through path when consecutive near-threshold events occur on the loop, thereby reducing the occurrence and length of zero-gain or near-zero-gain segments under surrogate-gradient BPTT. A persistent gradient explosion effect would require two conditions to hold simultaneously. First, the product of the loop weights must be sufficiently large, e.g., $|w_{C,M}w_{M,C}| > 1$. Second, the corresponding surrogate derivatives must remain active across consecutive time steps, meaning that both loop neurons repeatedly stay in the near-threshold regime. In practice, this risk can be mitigated by bounding the loop weights, for example by constraining $|w_{C,M}| \leq 1$ and $|w_{M,C}| \leq 1$, or by regularizing the local loop weights during training.

