# OpenReview forum: "LIF Recurrent Memory Enables Long-Horizon Spiking Computation"
_ICML.cc/2026/Conference — ICML 2026 regular_

### Official Review · Reviewer_jDuc · 2026-02-28

**Soundness:** 2
**Presentation:** 2
**Significance:** 2
**Originality:** 2
**Overall Recommendation:** 4
**Confidence:** 3

**Summary:**

This paper proposes LRMM, a recurrent spiking architecture built from four vanilla LIF neurons arranged in a local memory loop and modulated by three learned currents, aiming to address memory leakage and vanishing gradients in long-horizon sequence tasks without resorting to complex neuron models. The authors analyze a core problem in spiking RNNs—long-term information retention under BPTT—and the authors strive to analyze a central concept: how structured local recurrence can improve effective temporal gain through loop-induced coupling. Empirically, the model achieves competitive or superior performance on benchmarks such as PS-MNIST, SHD, SSC, and Binary Adding, while reporting significant theoretical energy advantages over LSTM and other recurrent SNN baselines.

**Compliance With Llm Reviewing Policy:**

Affirmed.

**Final Justification:**

I appreciate the authors’ effort in addressing my concerns during the rebuttal phase, especially the additional clarification and analysis on gradient stability. The provided upper-bound analysis helps justify that the proposed architecture avoids gradient explosion and offers some intuition on the role of the recurrent loop in stabilizing training.

However, I still remain somewhat conservative regarding the theoretical contribution. The current analysis primarily establishes a non-expansive upper bound on gradient propagation, but does not provide guarantees against vanishing gradients or long-term gradient preservation. **In particular, no non-trivial lower bound on multi-step gradient propagation is given.**  As a result, while the theory offers useful intuition, it does not fully support the stronger claims regarding long-horizon credit assignment.

That said, I acknowledge that the empirical results are strong and consistent across multiple benchmarks, and the proposed architectural design is both practical and relevant to the SNN setting. The rebuttal partially addressed my concerns by clarifying the intent of the analysis and strengthening the stability discussion, although it did not fundamentally resolve the limitations of the theoretical guarantees.

Overall, despite my reservations about the theoretical completeness, I find the empirical contributions and architectural insights to be valuable. I therefore slightly increase my score to 4, while **encouraging the authors to further strengthen the theoretical analysis (e.g., by analyzing whether gradients vanish over long horizons) in the final version.**

**Key Questions For Authors:**

See weaknesses.

**Limitations:**

No. The paper contains a brief impact statement but does not meaningfully discuss methodological limitations or broader societal implications. The authors are encouraged to explicitly acknowledge the current reliance on theoretical energy modeling, the absence of hardware deployment validation, and the scope limitations of benchmark evaluations. A short discussion of potential deployment contexts (e.g., edge devices, always-on sensing) and associated considerations would also improve transparency.

**Strengths And Weaknesses:**

**Originality:** The proposed LRMM computes three modulatory currents via fully connected transformations and nonlinear clamping (Eq.4–6), which functionally resemble input/forget/output gating in LSTM-style recurrent cells; thus, the method appears closer to a constrained gated-RNN reparameterization than a fundamentally new memory principle, and its distinction from prior gated RSNN variants could be clarified more explicitly.


**Soundness:** The stability analysis derives loop-induced one-step effective gains (Eq.19–21) but does not establish multi-step bounds or spectral stability guarantees, and the energy accounting (Section 4.4) models MAC/AC costs without explicitly incorporating the computational and communication overhead of the modulation function Φ (Eq.4–6), leaving system-level efficiency claims somewhat under-specified.


**Significance:** The paper addresses an important limitation of long-horizon learning in recurrent SNNs and demonstrates strong empirical results across benchmarks (Table 1); if the above concerns regarding novelty boundaries and implementation-level accounting are clarified, the work could have meaningful impact within the spiking/neuromorphic research community.

**Weakness 1: Conceptual Novelty Limited**

The LRMM computes three modulatory currents via fully connected transformations followed by nonlinear clamping (Eq.4–6), which functionally resemble input/forget/output gating mechanisms in LSTM-style recurrent cells. Although implemented using LIF micro-circuits, the overall behavior is conceptually close to a constrained gated recurrent cell rather than introducing a fundamentally new memory principle. The paper would benefit from a clearer delineation of novelty relative to existing gated RSNN variants or other structured recurrent designs.

**Weakness 2: Implementation–Efficiency Inconsistency**

While the paper emphasizes that each memory module consists of only four vanilla LIF neurons, the computation of the modulation function $\Phi$ requires additional fully connected layers and nonlinear operations (Eq.4–6). In neuromorphic implementations, such nonlinear transformations are non-trivial.

If $\Phi$ is shared across cells, it may introduce non-negligible communication and routing overhead due to broadcasting of modulatory signals. If $\Phi$ is instantiated per cell, the required computational and hardware resources could substantially exceed the claimed minimal design.

However, the energy analysis only accounts for MAC and AC operations and does not explicitly model communication or routing costs. Clarification on how $\Phi$ is realized in hardware, and how its system-level cost is incorporated into the efficiency claims, is necessary.

**Weakness 3: Theoretical Analysis Is Local and Surrogate-Dependent**

The theoretical section provides a one-step temporal gain analysis of BPTT and shows that the NM–NC loop introduces an additional coupling term in the local Jacobian. However, the analysis remains local and does not establish multi-step gradient bounds, spectral stability conditions, or guarantees against gradient explosion.

Furthermore, the derivations rely on surrogate gradients, and the relationship between the surrogate-based analysis and the true discrete spiking dynamics is not fully characterized.

As such, the theoretical contribution appears more explanatory than a rigorous stability result supporting long-horizon guarantees.

---

> ### Author Rebuttal · Authors · 2026-03-31
>
> ### W1: Conceptual Novelty Limited
> Thank you for this comment. LRMM is a novel SNN design that emphasize on achieving long term modeling with efficient and hardware friendly vanilla LIF neurons. The working mechanism is not simply a gating function because using the binary LIF neuron as gating can easily flush out existing memory. Therefore, other SNN works typically had to modify the neuron model for long sequence tasks.
>
> Here we engineered a unique memory loop with LIF neurons and demonstrated compelling results. Moreover, the LIF neuron relies on **additive** accumulation of input and internal state where they are treated equally without selective gating. This is in contrast with LSTM and GRU with **multiplicative** gating.
>
> ### W2: Implementation–Efficiency Inconsistency
>
> Thank you for this comment.
>
> 1. Cost of Eq. (4)–(6):
>
>     We would like to clarify the cost of Eq. (4) to Eq. (6). Eq. (4)–(6) does not imply three fully separate inference stages. During inference, we set m=1, so the modulation function in Eq. (5) reduces to the piecewise-linear clipping form in Eq. (6). The variable z in the clip operation is exactly the linear output computed in Eq. (4). Therefore, these equations are realized as one linear transform followed by one clip operation, rather than multiple independent nonlinear blocks.
>
>     In our current energy estimation, the cost of the linear computation in Eq. (4) has already been included in Table 8 of Appendix A.12.
>
> 2. Whether modulation is global or per-cell:
>
>     We would like to clarify that the modulation in LRMM is per-cell and feed to one neuron in its own loop.
>
> 3. Why the communication overhead is likely moderate:
>
>     Thank you for the comment. LRMM is a hierarchy of local-global recurrent connections.
>    We calculated the local and global spike generation data is shown in the table below (based on the data in Sec. 4.4, network size: 1044 neurons).
>     About 72% spikes occurred between $N_I$, $N_C$, and $N_M$ neurons are all local and do not incur core-to-core communications. Thus LRMM is much more efficient than standard RSNN with only global connections.
>
> |  | LRMM  | LIF | TC-LIF
> | - | -- | -- | -
> | Spikes per time step in SHD | 271.81 |257.70|276.08
> | local spikes per time step in SHD | 195.96 |0| 0
> | Spikes per sequence in SHD | 27181|25770|27608
>
> ### W3: Theoretical Analysis Is Local and Surrogate-Dependent
>
> Thank you very much for this insightful comment. Our method provides non negative temporal gain across multi-step by recurrent, but does establish a quantitative bound.
>
> Regarding gradient explosion, constraining the loop weights within a bounded range can help avoid excessive local gain, such as constraining $|w_{C,M}| \le 1,|w_{M,C}| \le 1.$
>
> In this work, we used well-established surrogate-based BPTT training framework for SNNs, but our key contribution is the design of a unique hierarchical recurrent structure that reduces temporal gradient vanishing in this widely used setting, as we discussed in Section 3.2.

---

> > ### Author Rebuttal · Reviewer_jDuc · 2026-04-01
> >
> > Thank you for the careful and detailed rebuttal. Your rebuttal provides helpful clarifications, particularly regarding the implementation and computational cost of the modulation function. These partially address some of the concerns. However, key issues, such as the conceptual novelty relative to gated recurrent models and the lack of stronger theoretical guarantees, remain only partially addressed.

---

> > > ### Author Response · Authors · 2026-04-07
> > >
> > > ### Q1. Novelty compared to LSTM or GRU like gated architecture.
> > >
> > > Thank you for the thoughtful follow-up and we have added further analysis of our LRMM architecture.
> > >
> > > 1) The target problem of this work is for a sparse SNN to maintain long sequence memory, which is not a problem in a dense LSTM network. Our contribution lies in the introduction of a local memory loop, in addition to the global recurrent connections, to effectively maintain long memory cues, this concept was not used in LSTM or GRU architecture and novel to the SNN designs.
> > >
> > > 2) We compared the results of LSTM, LSTM (convert gating to LIF), LRMM, and LRMM without the memory loop. It was found that neither direct conversion of LSTM to SNN, nor building SRNN without local memory loop work in our copy task experiment, as shown in table below.
> > >
> > > | Copy task    |  Acc   |
> > > |---------------|---
> > > | LRMM     | 99.58  |
> > > | LRMM w/o local loop |  71.49 |
> > > | LSTM(convert gating to LIF) | 76.83
> > > | LSTM | 99.43
> > >
> > > *Copy task: Input sequence containing signals in first 6 timesteps and repeating signals at the output after an interval of 20 timesteps. The step-wise accuracy of 6 classes of the input signals were reported.
> > >
> > > ### Q2:Theoretical Analysis Is Local
> > >
> > > **1. Gradient-backtracking measurement.**
> > >
> > > Thank you for this important comment. Here we further provide analysis of multistep gradient analysis as follow:
> > >
> > > For a fixed time point $\tau$, backpropagate from $L_\tau$, we calculate the mean gradient norm on internal states at time $\tau-H$, where $H\in\{5,10,15,20\}$.
> > >
> > > For LRMM, the mean is
> > > $$
> > > \text{grad}(H) = mean(\left\|\frac{\partial L_\tau}{\partial U_M(\tau-H)}\right\|,
> > > \left\|\frac{\partial L_\tau}{\partial U_C(\tau-H)}\right\|).
> > > $$
> > >
> > > For LSTM, the mean is
> > > $$
> > > \text{grad}(H) = mean(\left\|\frac{\partial L_\tau}{\partial h(\tau-H)}\right\|,
> > > \left\|\frac{\partial L_\tau}{\partial c(\tau-H)}\right\|).
> > > $$
> > >
> > > The relative value is normalized by the first valid horizon:
> > > $$
> > > \text{relative}(H)=\frac{\text{grad}(H)}{\text{grad}(5)}.
> > > $$
> > > Larger relative values indicate better multi-step gradient retention.
> > >
> > > Using these equations, we can measure the gradient of the loss at time $\tau$ with respect to the internal states $H$ steps earlier. We find that cutting the NM--NC loop leads to much faster gradient decay, while our LRMM architecture consistently shown stable gradient retention (without vanishing or explosion after 15 steps). This is consistent with our analysis: the loop helps preserve gradients over long horizons.
> > >
> > >
> > > | copy task gradient retention, relative to H =5 | H=5       | H=10      | H=15      | H=20      |
> > > |---------------|-----------|-----------|-----------|-----------|
> > > | LRMM  | 1.00     | 0.832    | 0.698    | 0.524    |
> > > | LRMM no-local loop | 1.00     | 0.056    | 1.92e-3    | 5.18e-6    |
> > > | LSTM  | 1.00     | 0.307  | 0.113 | 3.17e-2 |
> > > | LSTM to SNN  |1.00 | 0.162| 2.17e-2|3.04e-3
> > > *same task as above, using early-stage training data
> > >
> > >
> > > **2. We can avoid gradient explosion by constraining the loop weights $w_{C,M},w_{M,C}$ within a bounded range.**
> > >
> > > Under the box surrogate used in our paper (width =1), the surrogate derivative is nonzero only in the near-threshold support, and equals 1 on the active support. Hence
> > > $$
> > > A_X[t]=\alpha_X\bigl(1-V_{th}\sigma_X'(U_X[t]-V_{th})\bigr)
> > > $$
> > > has the following two cases:
> > >
> > > off threshold:  $A_X[t]=\alpha_X \le 1$;
> > > on the active support: $A_X[t]=0$.
> > >
> > > Therefore, in the direct-chain gains
> > > $$
> > > G_C[t]=A_C[t]+\beta_t\gamma_{t+1},\qquad
> > > G_M[t]=A_M[t]+\gamma_t\beta_t,
> > > $$
> > > the diagonal term $A_X[t]$ and the coupling correction do not contribute simultaneously at the same step.
> > >
> > > For $G_C[t],$ if $U_C[t]$ is off threshold, then $\beta_t=0$, hence
> > > $$
> > > G_C[t]=A_C[t]=\alpha_C\le 1.
> > > $$
> > > If $U_C[t]$ is on the active support, then $A_C[t]=0$, so
> > > $$
> > > |G_C[t]|=|\beta_t\gamma_{t+1}|.
> > > $$
> > > Since $|\sigma'(\cdot)|\le 1$, we have
> > > $$
> > > |\beta_t|\le |w_{CM}|,\qquad |\gamma_{t+1}|\le |w_{MC}|,
> > > $$
> > > thus
> > > $$
> > > |G_C[t]|
> > > \le |w_{CM}w_{MC}|.
> > > $$
> > > So if we constrain
> > > $$
> > > |w_{CM}w_{MC}|\le 1,
> > > $$
> > > then
> > > $$
> > > |G_C[t]|\le 1.
> > > $$
> > >
> > > The same argument gives
> > > $$
> > > |G_M[t]|\le 1.
> > > $$
> > >
> > > Hence, for the multi-step direct-chain coefficient
> > > $$
> > > \Gamma_X(t,H)=\prod_{k=0}^{H-1}G_X[t+k],
> > > $$
> > > we obtain
> > > $$
> > > |\Gamma_X(t,H)|
> > > \le
> > > \prod_{k=0}^{H-1}|G_X[t+k]|
> > > \le 1.
> > > $$
> > > Therefore, under the box surrogate (width =1) and bounded loop weights, the **chain is non-expansive** and does not cause gradient explosion.

---

### Official Review · Reviewer_RuSy · 2026-03-11

**Soundness:** 3
**Presentation:** 3
**Significance:** 3
**Originality:** 3
**Overall Recommendation:** 5
**Confidence:** 3

**Summary:**

The paper addresses a fundamental bottleneck in Spiking Neural Networks (SNNs): the trade-off between temporal memory and hardware efficiency. By proposing the LIF Recurrent Memory Module (LRMM), the authors shift the paradigm from designing complex, non-native neuron models to an architectural-level integration of vanilla LIF neurons. The manuscript is exceptionally well-written. The experimental section is comprehensive and robust.

**Compliance With Llm Reviewing Policy:**

Affirmed.

**Key Questions For Authors:**

1. Since the paper emphasizes an "architectural-level solution" over complex neuron models, can you clarify to what extent the performance gains are due to the specific local loop structure versus the increased parameter count?
2. In Section 3.2, you discuss how the loop-induced gain mitigates gradient disconnection. However, does this mechanism introduce a risk of gradient exploding, especially in very deep architectures or extremely long sequences?

**Limitations:**

yes

**Strengths And Weaknesses:**

Strengths:
1. Achieving 96.52% accuracy on PS-MNIST while maintaining 277x higher energy efficiency than LSTM is a standout result.
2. Demonstrating that LRMM can further benefit from hybrid designs (LRMM-ALIF) adds depth to the scalability analysis.
Weaknesses:
1. The stability analysis indicates that the loop-induced temporal gain mitigates gradient disconnection, but it does not provide a pointwise guarantee of nonzero gain at every single time step. Instead, it simply reduces the occurrence of zero-gain segments.
2. LRMM requires approximately 3× longer training time than a standard Recurrent Spiking Neural Network (RSNN). This is attributed to the additional computations introduced by its local recurrent modules.

---

> ### Author Rebuttal · Authors · 2026-03-31
>
> ### W1: The stability analysis indicates that the loop-induced temporal gain mitigates gradient disconnection, but it does not provide a pointwise guarantee of nonzero gain at every single time step. Instead, it simply reduces the occurrence of zero-gain segments.
>
> Thank you very much for this insightful comment. The reviewer is right that our analysis cannot guarantee nonzero temporal gain at every time step. Despite that, our method shows evident and significantly improvement in training. We will further explore combining it with other approaches to improve performance.
>
> ### W2: LRMM requires approximately 3× longer training time than a standard Recurrent Spiking Neural Network (RSNN). This is attributed to the additional computations introduced by its local recurrent modules.
>
> Thank you for pointing out this issue. Our model does require more training time than a standard LIF-RSNN due to more complex structure. However, this modification is NECESSARY to overcome the poor long sequence performance in standard LIF-RSNN (as shown in table below, for example **71.4% for LIF and 94.70% for LRMM on SHD**).
>
> Our method remains competitive in training time compared to related work. For example, TC-LIF is a representative work for long sequence tasks. Across all four tasks, LRMM is consistently faster and uses less GPU memory than TC-LIF.
>
>
> | Benchmark | LRMM Time | LRMM Acc (%) | LIF-RSNN Time | LIF-RSNN Acc (%) | TC-LIF Time | TC-LIF Acc (%) |
> |---|---|---:|---|---:|---|---:|
> | SHD | 41s | 94.70  | 16s | 71.40 | 51s |  88.91 |
> | PS-MNIST | 46min39s |  96.52  | 13min08s |  80.39 | 58min27s | 95.36 |
> | SSC | 8min32s | 79.75  | 3min20s | 50.90 | 8min58s  | 61.90 |
> | Binary Adding Problem | 35s |  99.55 | 12s | 53.35 | 39s |  19.90 |
>
> ### Q1: Since the paper emphasizes an "architectural-level solution" over complex neuron models, can you clarify to what extent the performance gains are due to the specific local loop structure versus the increased parameter count?
>
> Thank you for this important question.
> 1.  Our ablation study showed that when the NC to NM memory context loop is removed, the performance drops significantly, showing effectiveness of the local loop structure.
>
> 2.  We further added comparisons to standard LIF RSNN with same number of neurons (while LRMM has much smaller parameter), as shown in table below. LRMM has achieved consistent better results than LIF RSNN, indicating the importance of the local loop structure.
>
> |  | PS-MNIST acc | SSC acc | SHD acc| Binary adding acc| PS-MNIST params | SSC params | SHD params | Binary adding params |
> | - | -- | -- | - | - | -- | -- | -| -
> | LIF RSNN | 81.35% |54.97%|75.11%|58.45%|0.79M|0.86M|1.14M|0.79M
> | LRMM| 96.52%|79.75%|94.70%|99.55%|0.15M|0.20M|0.42M|0.15M
>
> *Both models have 2 layers and 512 LIF neurons per layer.
>
> ### Q2: In Section 3.2, you discuss how the loop-induced gain mitigates gradient disconnection. However, does this mechanism introduce a risk of gradient exploding, especially in very deep architectures or extremely long sequences?
>
> Thank you very much for this important question.
>
> Based on Eq. (19) to Eq. (21), a persistent exploding effect would require two conditions to hold at the same time. First, the loop gain must be sufficiently large, namely the product of the two loop weights must be greater than one in magnitude. Second, the corresponding surrogate derivatives must stay active across consecutive time steps, meaning that both neurons remain repeatedly in the near threshold regime.
>
> This risk can be controlled by constraining the loop weights within a bounded range, such as constraining $|w_{C,M}| \le 1, |w_{M,C}| \le 1$. We have clarified this point in the revised paper to better define the scope and stability boundary of the analysis.

---

> > ### Author Rebuttal · Reviewer_RuSy · 2026-04-01
> >
> > My concerns have been adequately addressed

---

> > > ### Author Response · Authors · 2026-04-02
> > >
> > > We sincerely thank you for the positive assessment and insightful comment. We truly appreciate your valuable time and encouraging feedback.

---

### Official Review · Reviewer_HJSg · 2026-03-13

**Soundness:** 2
**Presentation:** 3
**Significance:** 3
**Originality:** 3
**Overall Recommendation:** 4
**Confidence:** 4

**Summary:**

In this paper, the authors propose creating a "micro-circuit" of LIF neurons designed in a way that it can enhance the long-range information storage of the network. This micro-circuit is designed to stably propagate information forwards and gradients backwards. They test their method on various tasks and do a qualitative analysis of the networks gating and spiking mechanisms. They also provide an ablation study and estimated energy consumption of the network.

**Compliance With Llm Reviewing Policy:**

Affirmed.

**Final Justification:**

I have updated my score based on the author rebuttal, which addresses most of my major concerns.

**Key Questions For Authors:**

- Since this model now includes more neurons for a given task, how does this affect the memory access characteristics of networks compared to LSTM/EGRU?

**Limitations:**

Limitations haven't been discussed much.

**Strengths And Weaknesses:**

## Strengths

- The motivation behind the proposed method is very convincing. It makes a lot of sense to try to use (A)LIF neurons as building blocks to construct networks that can do better for efficiency reasons. This is very well articulated in the introduction (l. 55-l.65).
- The approach to my knowledge is very novel, and the authors use principled and sound reasoning to design the micro-circuit.
- The task performance on the benchmarks (esp. psMNIST) is very strong.
- The ablation study clearly shows the utility of each of the specific feature they include in their model.

## Weaknesses

- The empirical evaluation could be improved
  - The empirical comparison does not include standard SSM baselines on the shown tasks. (Also see bullet point regarding references).
  - Since the model promises improvement in long-range tasks, I would suggest also testing on standard benchmarks such as Long Range Arena or Language Modelling to show that it is not limited to spiking and small datasets.
- Many key references are missing:
  - col 2, l.39: The ALIF neuron was first introduced in [1].
  - col 2, l.67: The EGRU model [2] is a spiking gated recurrent model that is state-of-the-art in Language Modelling, and would be helpful to see how it compares to LRMM on task-performance and efficiency.
  - Similarly, the event-SSM [3] baseline for SHD & SSC would be helpful in showing where LRMM stands compared to hybrid models.

## Minor

- l. 330-350: The results would be a lot more readable in a table format rather than in text.
- Section 4.4:
  - The firing rates could be moved to the appendix to improve readability.
  - The paragraph on energy efficiency is repetitive -- most of the numbers mentioned there are already present in Table 3.
- Eq. (4): Not clear if the ';' was supposed to be multiplication.

### References

[1] Bellec, G., et al. (2018). Long short-term memory and Learning-to-learn in networks of spiking neurons. In Advances in Neural Information Processing Systems 31, S. Bengio, H. Wallach, H. Larochelle, K. Grauman, N. Cesa-Bianchi, and R. Garnett, eds. (Curran Associates, Inc.), pp. 787–797.

[2] Subramoney, A., et al. (2023). Efficient recurrent architectures through activity sparsity and sparse back-propagation through time. In The Eleventh International Conference on Learning Representations.

[3] Schöne, M., et al. (2024). Scalable Event-by-Event Processing of Neuromorphic Sensory Signals with Deep State-Space Models. In 2024 International Conference on Neuromorphic Systems (ICONS), pp. 124–131. https://doi.org/10.1109/ICONS62911.2024.00026.

---

> ### Author Rebuttal · Authors · 2026-03-31
>
> ### W1: Missing SSM baselines; we have added comparisons with SSM and other methods.
>
> 1. Thank you for this constructive suggestion. We have added comparisons with Event-SSM, and EGRU in the revised paper, as shown in table below.
>
> 2. LRMM has performed consistently better than EGRU on all three datasets, while achieves comparable performance to Event-SSM on SHD and PS-MNIST.
>
> 3. One notable difference is that our LRMM is a SNN designed with vanilla LIF neurons for sparse operation, while Event-SSM is an ANN with dense computations. From our evaluation, LRMM demonstrated much better power efficiencies over Event-SSM. On SHD dataset, energy per step per sample is 76.11 nJ for LRMM and 44,788 nJ for Event-SSM. **LRMM reduces the estimated energy cost by about  587×**.
>
> |  | LRMM  |  Event-SSM |EGRU
> | - | -- | ---|-
> | Spiking model | Yes|No|Yes
> | SSC Acc | 79.75%|88.4%|71.06%
> | Parameters on SSC | 0.20M|0.6M|2.31M
> | PS-MNIST Acc | 96.52%|97.17%|95.1%
> | Parameters on PS-MNIST | 0.15M|0.38M|1M
> | SHD Acc | 94.70%|95.9%|90.47%|
> | Parameters on SHD | 0.42M|0.4M|2.30M
> |Energy per step per sample on SHD|76.11nJ|44788nJ|2514nJ
>
> ### W2: I would suggest also testing on standard benchmarks such as Long Range Arena or Language Modelling to show that it is not limited to spiking and small datasets.
>
> Thank you for this suggestion. We agree that evaluating LRMM on more complex tasks would further demonstrate its robustness and generality. These tasks require substantial computing resources and long training period. We are actively working on new experiments to complement existing results.
>
> The LRMM presents a novel SNN architectural design that highlights the unique co-optimization of efficiency, accuracy and hardware compatibility of LRMM over these medium level tasks for use on edge computing.
>
> ### W3: Reference missing problem
>
> Thank you the suggestion, we have cited LSNN (Bellec et al. 2018), EGRU, and event-SSM in the revised paper, and we have also added experimental comparisons with EGRU and event-SSM.
>
> ### W4: Format problem.
> In the revised paper,
>
> 1. we have reduced repeated descriptions of numbers in the main text in l. 330-350.
>
> 2. We have moved detailed firing rate statistics to the appendix to keep the main text more readable in Sec 4.4.
>
> 3. We have also simplified the discussion of energy efficiency and avoided repeating values in Table 3.
>
> 4. Regarding Eq. (4), the symbol was intended to represent concatenation, and we have revised the notation and explanation to make this clear.
>
> ### Q1: How does the memory access pattern of LRMM compare with that of LSTM/EGRU?
>
> Thank you for the question.
> 1.  Memory access has two aspects in LRMM (neuronal and synaptic operations). The memory access is tightly associated with spike activity rather than static parameter count. For LSTM and EGRU, memory access is mainly dominated by dense matrix computation.
>
> 2.  We quantitatively analyze the per-step memory access, as shown in table below. Despite having more neuron groups, LRMM still incurs substantially lower total memory access than both LSTM and EGRU.
>
> |  | LRMM  | LSTM | EGRU
> | - | -- | -- | -
> | synaptic access per step in SHD | $6.24*10^4$ |$4.58*10^6$|$7.54*10^5$
> | neuron update | 1044 | 0 | 0
> |total memory access per step in SHD|$6.35*10^4$ |$4.58*10^6$|$7.54*10^5$

---

> > ### Author Rebuttal · Reviewer_HJSg · 2026-04-01
> >
> > Most of my concerns have been resolved. The testing on language modelling or long-range arena is pending. But it is a strong paper, and I will raise my score.

---

> > > ### Author Response · Authors · 2026-04-02
> > >
> > > We sincerely thank the reviewer for the thoughtful assessment and are grateful for your willingness to raise the score. We are very glad that most of your concerns have been resolved and you endorse this paper as a strong one. Any further comments or suggestions would be greatly appreciated.

---

### Official Review · Reviewer_eag4 · 2026-03-21

**Soundness:** 3
**Presentation:** 3
**Significance:** 2
**Originality:** 2
**Overall Recommendation:** 4
**Confidence:** 3

**Summary:**

In this paper the authors discuss how long sequences can be handled better without using complex neuron designs but instead small memory circuits out of just a few standard LIF neurons, and they argue that this little recurrent circuit helps both with remembering information over longer time spans and with keeping gradients from dying during training. Empirically, the method does really well on benchmarks like PS-MNIST, SHD, SSC, and Binary Adding, often beating more complicated baselines while also being more "energy efficient".

**Compliance With Llm Reviewing Policy:**

Affirmed.

**Ethical Review Concerns:**

No ethical concerns.

**Final Justification:**

Based on the clarifications from the rebuttal, I switched to weak accept.

**Key Questions For Authors:**

No questions.

**Limitations:**

The authors do not report limitations in depth beyond the extra time for training.

**Strengths And Weaknesses:**

Strengths
- Cool is that only LIF is used - and that a clever arrangement of LIF units can do the trick.
- Results seem very strong (see the validation issue below though).
- It is also nice that the authors try to explain why it works, with ablations and the gradient/memory-loop.
- The design is fairly interpretable for this kind of work, because the four parts have clear roles like input, memory, context, and readout.

Weaknesses
- Some evaluation details are vague, like whether results are from single runs or averaged over seeds? Top1 accuracy? Validation set used? Best test error achieved during a run (that would be sort of cheating)? Without this information the strength of the results is difficult to interpret.
- It is not easy to extract SOPs for comparison with other approaches. Since communication (physically moving the spikes) on chip is the most expensive part of compute it would be good to have an overview on how many spikes are generated for a sequence on average and also per time step.
- The increase in training time is quite high.
- Limitations are discussed only pretty lightly. The paper does not really go deep into failure cases, robustness issues, or where the method breaks.

---

> ### Author Rebuttal · Authors · 2026-03-31
>
> ### W1: The evaluation protocol is unclear, making the results hard to interpret and assess for reliability.
>
> Thank you for your constructive comment.
>
> 1. Our models have shown consistent training results with low variance. We have added statistical results over five runs with different random seeds. As shown in table below.
>
> 2. The evaluation metric is Top-1 accuracy.
>
> 3. We follow official dataset splits. For datasets with a validation split, we only used validation set during training. For datasets without a validation split, we trained without validation and only checked the result after the training completed.
>
> 4. We DID NOT use the test set for model selection at any stage, and we confirm that there is no test data leakage.
>
> The top-1 accuracy with mean acc and standard deviation are as follows:
> || PS-MNIST | SSC | SHD|Binary Adding
> |-|-:|-:|-:|-:
> | LRMM | 96.47±0.10 | 79.79±0.09 | 94.64±0.17 | 99.55±0.05
> | LRMM-ALIF | 97.35±0.11 | 80.54±0.13 | 95.23±0.20 |100.00±0.00
>
> We will revise in the final paper to include these details.
>
> ### W2: The communication cost in association with the average number of spikes per sequence and per time step for comparison with prior methods.
>
> Thank you for this helpful suggestion. The spike generation data is added as follows (based on the data in Sec. 4.4, network size: 1044 neurons):
>
> |  | LRMM  | LIF | TC-LIF
> | - | -- | -- | -
> | Spikes per time step in SHD | 271.81 |257.70|276.08
> | local spikes per time step in SHD | 195.96 |0| 0
> | Spikes per sequence in SHD | 27181|25770|27608
>
> 1. Our model shows similar spike count per sequence and per time step to LIF and TC-LIF models with same number of neurons.
>
> 2. LRMM is a hierarchy of local-global recurrent connections. About 72% spikes occurred between $N_I$, $N_C$, and $N_M$ neurons which are all local and do not incur core-to-core communications. Thus LRMM is much more efficient than LIF and TC-LIF based RNNs with only global connections.
>
> We will add these statistics and clarify this distinction in the final paper.
>
> ### W3: The increase in training time is quite high.
>
> Thank you for pointing out this issue. Our model does require more training time than a standard LIF-RSNN due to more complex structure. However, this modification is NECESSARY to overcome the poor long sequence performance in standard LIF-RSNN (as shown in table below, for example, **71.4% for LIF and 94.70% for LRMM on SHD**).
>
> Our method remains competitive in training time compared to related work. For example, TC-LIF is a representative work for long sequence tasks. Across all four tasks, LRMM is consistently faster and uses less GPU memory than TC-LIF.
>
> | Dataset | LRMM Time | LIF-RSNN Time | TC-LIF Time  | LRMM Mem (MB) | LIF-RSNN Mem (MB) | TC-LIF Mem (MB) | LRMM Acc (%) | LIF-RSNN Acc (%) | TC-LIF Acc (%) |
> | - | - | - | - | -: | -: | -: | -: | -: | -: |
> | SHD | 41s | 16s | 51s | 754 | 536 | 1250 | 94.64 ± 0.17 | 71.40 | 88.91 |
> | PS-MNIST | 46min39s | 13min08s | 58min27s | 3144 | 1374 | 4480 | 96.47 ± 0.10 | 80.39 | 95.36 |
> | SSC | 8min32s | 3min20s | 8min58s | 750 | 532 | 1246 | 79.79 ± 0.09 | 50.90 | 61.90 |
> | Binary Adding Problem | 35s | 12s | 39s | 714 | 508 | 1182 | 99.55 ± 0.05 | 53.35 | 19.90 |
>
> *All models have 2 layers and 512 neurons per layer.
>
> ### W4: Limitations are discussed lightly.
>
> Thank you for this comment. We have expanded the limitation discussion and will update it in the final manuscript.
>
> - **Long sequence**: Our current study has demonstrated unique  co-optimization of accuracy, efficiency and hardware compatibility in medium scale sequence benchmarks. However, exploring the feasibility of LRMM in ultra-long tasks, such as long context reasoning and decision making, is also interesting and important. Extending LRMM to these more demanding long-sequence settings remains an important direction for future work.
>
> - **Hardware**: The efficiency of SNNs will be fully exploited when deployed in specialized neuromorphic hardware. Practical hardware deployment introduce additional model constraints, such as limited weight precision, routing overhead, core mapping constraints, have not been optimized by our current evaluation. Meanwhile, specific hardware optimization for local recurrent circuits in LRMM may further improve the processing efficiency in hardware.
>
> - **Cross-architecture**: LRMM works well in current sequence modeling tasks with medium network size, while its scalability to large network size, such as using a hierarchical design for more complex memory tasks, as well as integration with other network architectures, such as transformers, have not been explored. Such cross-architecture design require substantial theoretical innovations in the future work.

---

> > ### Author Rebuttal · Reviewer_eag4 · 2026-04-06
> >
> > The rebuttal makes things a bit clearer now. Many thanks.

---

> > > ### Author Response · Authors · 2026-04-07
> > >
> > > We are grateful for the reviewer’s positive response and kind consideration in raising the score. We are very pleased that your concerns have been resolved. Any further comments or suggestions would be highly appreciated.

---

### Decision · Program_Chairs · 2026-04-30

**Decision:**

Accept (regular)

**Comment:**

This paper proposes LRMM, a recurrent memory module built from vanilla LIF neurons for long-horizon spiking computation. Overall, the reviewers found the problem important, the empirical results strong, and the architectural idea relevant to efficient spiking/neuromorphic sequence modeling. They also appreciated the ablations and mechanistic analyses.

The main concerns were consistent across reviews: the theory remains largely local and explanatory rather than providing strong long-horizon guarantees; the novelty boundary relative to gated recurrent designs should be clarified more carefully; and the efficiency/evaluation discussion could be broader, including clearer protocol details, stronger baselines, and more explicit accounting of implementation-level overheads.

In rebuttal, the authors addressed many of these issues constructively by adding multi-seed results, clarifying the evaluation setup, including additional baselines, expanding the efficiency and memory-access discussion, and further explaining gradient behavior and the role of the local memory loop. These responses were received positively overall: one reviewer considered the concerns fully resolved, and three reviewers raised scores to weak accept. At the same time, three reviewers remained somewhat cautious, especially about theoretical analysis, while finding the empirical contributions and architectural insights to be valuable.

Taking the reviews and rebuttal together, my assessment is comparatively positive. The final version should better calibrate the theoretical claims. My recommendation is weak accept.